# ECLipsE: Efficient Compositional Lipschitz Constant Estimation for Deep Neural Networks

**Yuezhu Xu**
Edwardson School of Industrial Engineering
Purdue University
West Lafayette, IN, USA
xu1732@purdue.edu

**S. Sivaranjani**
Edwardson School of Industrial Engineering
Purdue University
West Lafayette, IN, USA
sseetha@purdue.edu

## Abstract

The Lipschitz constant plays a crucial role in certifying the robustness of neural networks to input perturbations. Since calculating the exact Lipschitz constant is NP-hard, efforts have been made to obtain tight upper bounds on the Lipschitz constant. Typically, this involves solving a large matrix verification problem, the computational cost of which grows significantly for both deeper and wider networks. In this paper, we provide a compositional approach to estimate Lipschitz constants for deep feed-forward neural networks. We first obtain an *exact* decomposition of the large matrix verification problem into smaller sub-problems. Then, leveraging the underlying cascade structure of the network, we develop two algorithms. The first algorithm explores the geometric features of the problem and enables us to provide Lipschitz estimates that are comparable to existing methods by solving small semidefinite programs (SDPs) that are only as large as the size of each layer. The second algorithm relaxes these sub-problems and provides a closed-form solution to each sub-problem for extremely fast estimation, altogether eliminating the need to solve SDPs. The two algorithms represent different levels of trade-offs between efficiency and accuracy. Finally, we demonstrate that our approach provides a steep reduction in computation time (as much as several thousand times faster, depending on the algorithm for deeper networks) while yielding Lipschitz bounds that are very close to or even better than those achieved by state-of-the-art approaches in a broad range of experiments*. In summary, our approach considerably advances the scalability and efficiency of certifying neural network robustness, making it particularly attractive for online learning tasks.

## 1 Introduction

The Lipschitz constant, which quantifies how a neural networks output varies in response to changes in its inputs, is a crucial measure in providing robustness certificates [1, 2] on downstream tasks such as ensuring resilience against adversarial attacks [3, 4], stability of learning-based models or systems with neural network controllers [5–9], enhancing generalizability [10], improving gradient-based optimization methods and controlling the rate of learning [11][12]. The problem of calculating the exact Lipschitz constant is NP-hard [13]. Therefore, efforts have been made to estimate tight upper bounds for the Lipschitz constant of feed-forward neural networks (FNNs) [14–18] and other architectures such as convolutional neural networks (CNNs) [19–21]. Typical approaches include formulating a polynomial optimization problem [22] or bounding the Lipschitz constant via quadratic constraints and semidefinite programming (SDP) [14], which in turn requires solving a large-scale matrix verification problem whose computational complexity grows significantly with

---

*https://github.com/YuezhuXu/ECLipsE

both the depth and width of the network. These approaches have also motivated the development of methods to design neural networks with certifiable robustness guarantees [19, 23–25].

**Contribution.** In this paper, we provide a scalable compositional approach to estimate Lipschitz constants for deep feed-forward neural networks. We demonstrate steep reductions in computation time (as much as several thousand times faster than the state-of-the-art depending on the experiment), while obtaining Lipschitz estimates that are very close to or even better than those achieved by state-of-the-art approaches. Specifically, we develop two algorithms, representing different levels in the trade-off between accuracy and efficiency, allowing for application-specific choices. The first algorithm, `ECLipsE`, involves estimating the Lipschitz constant through a compositional layer-by-layer solution of small SDPs that are only as large as the weight matrix in each layer. The second algorithm, `ECLipsE-Fast`, provides a ***closed-form solution*** to estimate the Lipschitz constant, completely eliminating the need to solve any matrix inequality SDPs. Both algorithms provably guarantee the existence of solutions at each step to generate tight Lipschitz estimates. In summary, our work significantly advances scalability and efficiency in certifying neural network robustness, making it applicable to a variety of online learning tasks.

**Theoretical Approach.** We begin with the large matrix verification SDP for Lipschitz constant estimation under the well-known framework LipSDP [14]. To avoid handling a large matrix inequality, we employ a sequential Cholesky decomposition technique to obtain an ***exact*** decomposition of the large matrix verification problem into a series of smaller, more manageable sub-problems that are only as large as the size of the weight matrix in each layer. Then, observing the cascade structure of the neural network, we develop (i) algorithm `ECLipsE`, which characterizes the geometric features of the optimization problem and enables us to provide an accurate Lipschitz estimate and (ii) algorithm `ECLipsE-Fast`, which further relaxes the sub-problems, and yields a closed-form solution for each sub-problem that altogether eliminates the need to solve any SDPs, resulting in extremely fast implementations.

**Related Work.** The simplest way to estimate the Lipschitz constans is to provide a naive upper bound using the product of induced weight norms, which is rather conservative [26]. Another approach is to utilize automatic differentiation to approximate a bound, which is not a strict upper bound, although it is often so in practice [13]. Additionally, compositions of nonexpansive averaged operators and affine operators [16], Clarke Jacobian based approaches and other methods focusing on local Lipschitz constants [17][27] have also been studied. Recently, optimization-based approaches such as sparse polynomial optimization [22] and SDP methods such as the canonical LipSDP framework [14] have been successful in providing tighter Lipschitz bounds. SDP-based methods specifically exploit the slope-restrictedness of the activation functions to cast the problem of estimating a Lipschitz constant as a linear matrix verification problem. However, the computational cost of such methods explodes as the number of layers increases. A common strategy to address this is to ignore some coupling constraints among the neurons to reduce the number of decision variables, yielding a more scalable algorithm at the expense of estimation accuracy [14]. Another strategy is to exploit the sparsity of the SDP using graph-theoretic approaches to decompose it into smaller linear matrix inequalities (LMI) [15][28]. Along similar lines, [21] and [29] employ a dissipativity-based method and dynamic convolutional partition respectively to derive layer-wise LMIs that are applicable to both FNNs and CNNs. Very recent developments also focus on enhancing the scalability of SDP-based implementations through eigenvalue optimization and memory improvement [20], which are compatible with autodiff frameworks such as PyTorch and TensorFlow.

## 2 Problem Formulation and Background

**Notation.** We define $\mathbb{Z}_N = \{1, \ldots, N\}$, where $N$ is a natural number excluding zero. A symmetric positive-definite matrix $P \in \mathbb{R}^{n \times n}$ is represented as $P > 0$ (and as $P \geq 0$, if it is positive semi-definite). We denote the largest singular value or the spectral norm of matrix $A$ by $\sigma_{max}(A)$. The set of positive semi-definite diagonal matrices is written as $\mathbb{D}_+$.

### 2.1 Problem Formulation

We consider a feedforward neural network (FNN) of $l$ layers with input $z \in \mathbb{R}^{d_0}$ and output $y \in \mathbb{R}^{d_l}$ defined as $y = f(z)$. The function $f$ is recursively formulated with layers $\mathbf{L}_i, i \in \mathbb{Z}_l$, defined as

$$\mathbf{L}_i : z^{(i)} = \phi(v^{(i)}) \quad \forall i \in \mathbb{Z}_{l-1}, \quad \mathbf{L}_l : y = f(z) = z^{(l)} = v^{(l)}, \quad z^{(0)} = z, \qquad (1)$$

where $v^{(i)} = W_i z^{(i-1)} + b_i$ with $W_i$ and $b_i$ representing the weight and bias for layer $\mathbf{L}_i$ respectively, and $\phi : \mathbb{R}^{d_i} \to \mathbb{R}^{d_i}$ is a nonlinear *activation function* that acts element-wise on its argument. The last layer $\mathbf{L}_l$ is termed the *output layer*. We denote the number of neurons in layer $\mathbf{L}_i$ by $d_i$, $i \in \mathbb{Z}_l$.

**Definition 1.** *A function $f : \mathbb{R}^{d_0} \to \mathbb{R}^{d_l}$ is Lipschitz continuous on $\mathcal{Z} \subseteq \mathbb{R}^{d_0}$ if there exists a constant $L > 0$ such that $\|f(z_1) - f(z_2)\|_2 \le L\|z_1 - z_2\|_2, \forall z_1, z_2 \in \mathcal{Z}$. The smallest positive $L$ satisfying this inequality is termed the Lipschitz constant of the function $f$.*

Without loss of generality, we assume $W_i \ne 0$, $i \in \mathbf{Z}_l$, as any weights being 0 will lead to the trivial case where the output corresponding to any input will remain the same after that layer. Our goal is to provide a scalable approach to give an efficient and accurate upper bound for the Lipschitz constant $L > 0$. Note that the proofs of all the theoretical results in this paper are included in Appendix A.

## 2.2 Preliminaries

We begin with a slope-restrictedness property satisfied by most activation functions, which is typically leveraged to to derive SDPs for Lipschitz certificates [14].

**Assumption 1** (Slope-restrictedness). *For the neural network defined in* (1)*, the activation function $\phi$ is slope-restricted in $[\alpha, \beta]$, $\alpha < \beta$ in the sense that $\forall v_1, v_2 \in \mathbb{R}^n$, we have $\alpha(v_1 - v_2) \le \phi(v_1) - \phi(v_2) \le \beta(v_1 - v_2)$ element-wise. Consequently, we have that for $\forall \Lambda \in \mathbb{D}_+$,*

$$\begin{bmatrix} v_1 - v_2 \\ \phi(v_1) - \phi(v_2) \end{bmatrix}^T \begin{bmatrix} p\Lambda & -m\Lambda \\ -m\Lambda & \Lambda \end{bmatrix} \begin{bmatrix} v_1 - v_2 \\ \phi(v_1) - \phi(v_2) \end{bmatrix} \le 0, \quad p = \alpha\beta, \quad m = (\alpha + \beta)/2. \quad (2)$$

Now, we can obtain an upper bound for the Lipschitz constant as follows; this result is equivalent to the well-known LipSDP framework [14].

**Theorem 1** (LipSDP). *For the FNN* (1) *satisfying Assumption 1, if there exists $F > 0$ and positive diagonal matrices $\Lambda_i \in \mathbb{D}_+$, $i \in \mathbb{Z}_{l-1}$ such that with $p = \alpha\beta$ and $m = \frac{\alpha+\beta}{2}$,*

$$\begin{bmatrix} I + pW_1^T\Lambda_1 W_1 & -mW_1^T\Lambda_1 & 0 & \dots & 0 \\ -m\Lambda_1 W_1 & \Lambda_1 + pW_2^T\Lambda_2 W_2 & -mW_2^T\Lambda_2 & \dots & 0 \\ 0 & -m\Lambda_2 W_2 & \Lambda_2 + pW_3^T\Lambda_3 W_3 & \dots & 0 \\ & & \vdots & & \\ 0 & \dots & -m\Lambda_{l-2}W_{l-2} & \Lambda_{l-2} + pW_{l-1}^T\Lambda_{l-1}W_{l-1} & -mW_{l-1}^T\Lambda_{l-1} \\ 0 & 0 & \dots & -m\Lambda_{l-1}W_{l-1} & \Lambda_{l-1} - FW_l^T W_l \end{bmatrix} > 0, \quad (3)$$

*then $\left\| z_2^{(l)} - z_1^{(l)} \right\|_2 \le \sqrt{1/F} \left\| z_2^{(0)} - z_1^{(0)} \right\|_2$, which provides a sufficient condition for the Lipschitz constant $L$ to be upper bounded by $\sqrt{1/F}$.*

***Remark*** 1. LipSDP provides three variants that tradeoff accuracy and efficiency, namely, LipSDP-Network, LipSDP-Neuron, and LipSDP-Layer, whose scalability increases sequentially at the expense of decreased accuracy. However, [30] provides a counterexample showing that the Lipschitz estimate from LipSDP-Network is not a strict upper bound; thus, only LipSDP-Neuron, and LipSDP-Layer are valid. Theorem 1 here directly corresponds to LipSDP-Neuron. If all $\Lambda_i$, $i \in \mathbb{Z}_{l-1}$ in (3) are set to multiples of identity matrices, that is, $\lambda_i I$, $i \in \mathbb{Z}_{l-1}$, then it corresponds to LipSDP-Layer.

Assumption 1 holds for all commonly used activation functions; for example, it holds with $\alpha = 0$, $\beta = 1$, that is, $p = 0, m = 1/2$ for the ReLU, sigmoid, tanh, exponential linear functions. Therefore, we focus on this case in this work.

## 3 Methodology

We now develop two fast compositional algorithms based on LipSDP-Layer and Lipschitz-Neuron respectively. Both algorithms are not only scalable and significantly faster, but also provide comparable estimates for the Lipschitz constant.

### 3.1 Exact Decomposition

We circumvent direct solution of the large matrix inequality in (3), which becomes computationally prohibitive as the FNN (1) grows deeper. Instead, we develop a sequential block Cholesky decomposition method, akin to the technique introduced in [31], also expanded in [32, 33]. We first restate Lemma 2 of [31] below.

**Theorem 2** (Restatement of Lemma 2 of [31])**.** *A symmetric block tri-diagonal matrix defined as*

$$\begin{bmatrix} \mathcal{P}_1 & \mathcal{R}_2 & 0 & \dots & & 0 \\ \mathcal{R}_2^T & \mathcal{P}_2 & \mathcal{R}_3 & \dots & & 0 \\ 0 & \mathcal{R}_3^T & \mathcal{P}_2 & \mathcal{R}_3 & \dots & 0 \\ & & \vdots & & & \\ 0 & \dots & 0 & \mathcal{R}_{l-1}^T & \mathcal{P}_{l-1} & \mathcal{R}_l \\ 0 & \dots & & 0 & \mathcal{R}_l^T & \mathcal{P}_l \end{bmatrix}, \tag{4}$$

*is positive definite if and only if $X_i > 0, \forall i \in \{0\} \cup \mathbb{Z}_{l-1}$, where*

$$X_i = \begin{cases} \mathcal{P}_i & \text{if } i = 0, \\ \mathcal{P}_i - \mathcal{R}_i^T X_{i-1}^{-1} \mathcal{R}_i & \text{if } i \in \mathbb{Z}_{l-1}. \end{cases} \tag{5}$$

**Theorem 3.** *Let $P_l$ be defined as in (3) with $p = 0, m = 1/2$. Then, the Lipschitz certificate $P_l > 0$ holds if and only if the following sequence of matrix inequalities is satisfied:*

$$M_i > 0, \quad \forall i \in \mathbb{Z}_{l-2}, \qquad M_{l-1} - FW_l^T W_l > 0, \tag{6}$$

*where*

$$M_i = \begin{cases} I & i = 0 \\ \Lambda_i - \frac{1}{4}\Lambda_i W_i (M_{i-1})_l^{-1} W_i^T \Lambda_i & i \in \mathbb{Z}_{l-1} \end{cases}. \tag{7}$$

Theorem 3 provides an **exact decomposition** of (3), and allows us to establish necessary and sufficient conditions through small matrix inequalities that scale with the size of the weight matrices of each layer, rather than that of the entire network. To accurately estimate the Lipschitz constant, we need to decide on $\Lambda_i, i \in \mathbb{Z}_{1-1}$ that generate a tight upper bound at the last stage. In other words, we want $M_{l-1} - FW_l^T W_l > 0$ to yield the smallest estimate for $\sqrt{1/F}$. In the following subsection, we provide compositional algorithms to decide the appropriate $\Lambda_i, i \in \mathbb{Z}_{1-1}$ sequentially, so that we only need to solve one small problem corresponding to each layer.

## 3.2 Compositional Algorithms

We first propose two practical algorithms here. The theory supporting the algorithms and the geometric intuition are deliberately deferred, and will be thoroughly discussed in a the next subsection.

The first algorithm, **ECLipsE**, explores the geometric features that enables us to provide an accurate Lipschitz estimate by solving small semidefinite programs (SDPs), which are of the size of the weight matrices on each layer. The second algorithm, **ECLipsE-Fast** relaxes the sub-problems at each stage and yields a closed-form solution for each sub-problem that makes it extremely fast. These algorithms represent different trade-offs between efficiency and accuracy; one may choose **ECLipsE** if pursuing accuracy, and **ECLipsE-Fast** for applications where time is of the essence.

We observe in (7) that $M_i$ is obtained in a recursive manner and depends on $\Lambda_i$ and $M_{i-1}, i \in \mathbb{Z}_{l-1}$. Therefore, we decide $\Lambda_i$ and then calculate $M_i$ for $i \in \mathbb{Z}_{l-1}$ sequentially. Thus, these two algorithms can be implemented layer-by-layer in a compositional manner.

Concretely, for **ECLipsE**, we obtain $\Lambda_i, i \in \mathbb{Z}_{l-1}$ at each stage $i$ using the information from the next layer, i.e. $W_{i+1}$, by solving the following ***small SDP***:

$$\max_{c_i} c_i \quad \text{s.t.} \begin{bmatrix} \Lambda_i - c_i W_{i+1}^T W_{i+1} & \frac{1}{2}\Lambda_i (W_i (M_{i-1})^{-1} W_i^T)^{\frac{1}{2}} \\ \frac{1}{2}(W_i (M_{i-1})^{-1} W_i^T)^{\frac{1}{2}} \Lambda_i & I \end{bmatrix} > 0, \ \Lambda_i \in \mathbb{D}_+, \ c_i > 0 \tag{8}$$

For **ECLipsE-Fast**, $\Lambda_i$ is reduced to $\lambda_i I, i \in \mathbb{Z}_{l-1}$ and $\lambda_i$ is calculated in ***closed-form*** as

$$\lambda_i = \frac{2}{\sigma_{max}\left(W_i (M_{i-1})^{-1} W_i^T\right)}. \tag{9}$$

Note that this ***completely eliminates*** the need to solve matrix inequality ***SDPs*** altogether. At last, after all $\Lambda_i$s, $i \in \mathbb{Z}_{l-1}$ are decided, we obtain the smallest $1/F$, which yields the smallest Lipschitz estimate $L = \sqrt{1/F}$, as follows

$$1/F = \sigma_{max}\left(W_l^T W_l (M_{l-1})^{-1}\right). \tag{10}$$

*Remark* 2. We choose to directly calculate the smallest $1/F$ rather than first derive the largest $F$. This is because obtaining the largest $F$ first involves taking the inverse of $W_l^T W_l$, which can cause numerical issues due to potential singularity of $W_l^T W_l$. In contrast, directly calculating the smallest $1/F$ involves taking the inverses of $M_{l-1}$, which is already guaranteed to be strictly positive definite at layer $l-1$ when deciding $\Lambda_{l-1}$.

We summarize the algorithms as one in Algorithm 1. Algorithms **ECLipsE** and **ECLipsE-Fast** are respectively preferable based on whether the priority is on accuracy or speed.

---

**Algorithm 1 ECLipsE** and **ECLipsE-Fast**

---

**Input** Weights $\{W_i\}_{i=1}^l$ from a FNN (1) with activation function slope-restricted in $[0,1]$
**Output** Lipschitz estimate $L$
1: Set $M_0 = I$
2: **for** $i = 1, 2, ..., l-1$ **do**
3:     **if ECLipsE** (pursuing accuracy) **then**
4:        Obtain $\Lambda_i$ from the optimal solution of (8)
5:     **else if ECLipsE-Fast** (pursuing speed) **then**
6:        Obtain $\lambda_i$ from (9)
7:        $\Lambda_i \leftarrow \lambda_i I$
8:     **end if**
9:     Obtain $M_i$ from (7) with $\Lambda_i$ and $M_{i-1}$
10: **end for**
11: Obtain $1/F$ from (10)
12: **Return** $L = \sqrt{1/F}$

---

### 3.3 Theory

Now we dive into the cascade structure of feed-forward neural networks and demonstrate the theory behind the two algorithms. We analyze the compositional algorithms in Section 3.2 in a backward manner, starting with the output layer. After all $\Lambda_i$, $i \in \mathbb{Z}_{l-1}$ are decided, $M_i > 0$, $i \in \mathbb{Z}_{l-2}$ hold. From Theorem 3, it remains to guarantee that $M_{l-1} - F W_l^T W_l > 0$, and consequently, (10), for which we state the following result.

**Proposition 1.** *For given $\Lambda_i$, $i \in \mathbb{Z}_{l-1}$ that satisfies $M_i > 0$, $i \in \mathbb{Z}_{l-2}$, the tightest upper bound for Lipschitz constant is $L = \sqrt{\sigma_{max}\left(W_l^T W_l (M_{l-1})^{-1}\right)}$.*

Now, at stage $l-1$, when deciding $\Lambda_{l-1}$, $\Lambda_i$, $i \in \mathbb{Z}_{l-2}$ are fixed and thus $M_{l-2}$ is fixed. According to Proposition 1, we would like to choose $\Lambda_{l-1}$ such that $\sigma_{max}\left(W_l^T W_l (M_{l-1})^{-1}\right)$, where $M_{l-1}$ is a function of $\Lambda_{l-1}$, is as small as possible. We have the following result.

**Lemma 1.** *If $M_i > 0$, then $W_{i+1}^T W_{i+1}(M_i)^{-1}$ and $W_{i+1}(M_i)^{-1}(W_{i+1})^T$ share the same non-zero eigenvalues.*

Note that at stage $i$, it is guaranteed that $M_i > 0$. Taking $i = l-1$, Lemma 1 infers that it is equivalent to minimize $\sigma_{max}\left(W_l(M_{l-1})^{-1}W_l^T\right)$ when deciding on $\Lambda_{l-1}$. Note that $M_{l-1} > 0$, and consequently, the existence of $M_{l-1}^{-1}$ is already guaranteed when we reach the last stage. For the sake of conciseness, we define $\mathcal{F}_i \triangleq W_i(M_{i-1})^{-1}W_i^T$   $i \in \mathbb{Z}_{l-1}$. From (7), $M_i = \Lambda_i - \frac{1}{4}\Lambda_i \mathcal{F}_i \Lambda_i$. We further write out the recursive expression for $\mathcal{F}_i$ as

$$\mathcal{F}_{i+1} = W_{i+1}(M_i)^{-1}W_{i+1}^T = \begin{cases} W_1 W_1^T & i = 0 \\ W_{i+1}(\Lambda_i - \frac{1}{4}\Lambda_i \mathcal{F}_i \Lambda_i)^{-1}W_{i+1}^T & i \in \mathbb{Z}_{l-1} \end{cases}. \quad (11)$$

**Lemma 2.** *For any constant $\gamma \in (0,1)$, any $\Lambda_i \in \mathbb{D}_+$ that satisfies $M_i = \Lambda_i - \frac{1}{4}\Lambda_i \mathcal{F}_i \Lambda_i > 0$ is also a feasible solution for $\tilde{M}_i \triangleq \Lambda_i - \frac{1}{4}\Lambda_i(\gamma \mathcal{F}_i)\Lambda_i > 0$. In other words, the feasible region $\{\Lambda_i : M_i > 0, \Lambda_i \in \mathbb{D}_+\} \subseteq \{\Lambda_i : \tilde{M}_i > 0, \Lambda_i \in \mathbb{D}_+\}$.*

Lemma 2 gives us the observation that a contraction $\mathcal{F}_i \to \gamma \mathcal{F}_i, \gamma \in (0,1)$ yields a larger feasible space for $\Lambda_i \in \mathbb{D}_+$ to ensure $M_i > 0$. Meanwhile, (11) shows that for any given $\Lambda_i$, a

smaller $\mathcal{F}_i$ leads to a smaller $\mathcal{F}_{i+1}$ for the next stage. We can characterize how 'small' $\mathcal{F}_i$ is by its spectral norm $\sigma_{max}(\mathcal{F}_i)$. Then, minimizing $\sigma_{max}(\mathcal{F}_i)$ aligns with our goal of minimizing $\sigma_{max}\left(W_l^T W_l (M_{l-1})^{-1}\right) = \sigma_{max}\left(W_l(M_{l-1})^{-1}W_l^T\right) = \sigma_{max}(\mathcal{F}_l)$ at the last stage. In other words, a smaller $\mathcal{F}_1$ at the start will generally translate to a tighter Lipschitz estimate at output layer if we always choose to minimize the spectral norm $\sigma_{max}(F_i)$ at each stage.

Now we focus on how to specifically optimize $\Lambda_i$, $i \in \mathbb{Z}_{l-1}$. At stage $i$, the goal is to seek for the $\Lambda_i$ that minimizes $\sigma_{max}(\mathcal{F}_{i+1})$, where $\mathcal{F}_{i+1} = W_{i+1}(\Lambda_i - \frac{1}{4}\Lambda_i \mathcal{F}_i \Lambda_i)^{-1} W_{i+1}^T$ as in (11). Note that $M_{i-1}$ and $\mathcal{F}_i$ are already fixed and can be regarded as constants at the $i$-th stage.

**Proposition 2.** *If there exists a singular matrix $N \geq 0$ such that $M_i = c_i W_{i+1}^T W_{i+1} + N$, with constant $c_i > 0$, then $\sigma_{max}(\mathcal{F}_{i+1}) = 1/c_i$, $\forall i \in \mathbb{Z}_{l-1}$.*

In other words, we need to find the largest $c_i > 0$ to minimize $\sigma_{max}(\mathcal{F}_{i+1}) = 1/c_i$. Recall that $M_i = \Lambda_i - \frac{1}{4}\Lambda_i \mathcal{F}_i \Lambda_i$ is a function of $\Lambda_i$. We state the following proposition that is used to derive the small sub-problems at each stage.

**Proposition 3.** *Consider the following optimization problem for $\forall i \in \mathbb{Z}_{l-1}$.*

$$\max_{c_i} \quad c_i \quad s.t. \quad \Lambda_i - \frac{1}{4}\Lambda_i W_i (M_{i-1})^{-1} W_i^T \Lambda_i - c_i(W_{i+1}^T W_{i+1}) > 0, \quad \Lambda_i \in \mathbb{D}_+, \quad c_i > 0 \tag{12}$$

*Then, the optimal value $c_i$ is the largest constant such that $M_i$ can be written as $M_i = c_i W_{i+1}^T W_{i+1} + N$, where $N$ is some singular matrix such that $N \geq 0$. Moreover, the feasible region for the optimization problem is always nonempty.*

***Geometric Analysis:*** We illustrate the process of achieving the largest $c_i > 0$ in Fig. 1. We geometrically represent a positive semidefinite matrix by the ellipsoid generated by the transformation of a unit ball in the Euclidean space by the matrix. For simplicity of exposition, we refer to this ellipsoid as the 'shape' of the matrix. We plot the shapes of $M_i$ and $W_{i+1}^T W_{i+1}$ in green and blue, respectively, in 2D. The positive definiteness of the constraint in (12) is equivalent to the ellipsoid of $W_{i+1}^T W_{i+1}$ being contained in the ellipsoid corresponding to $M_i/c_i$. Specifically, when $c_i > 1$, Fig. 1a demonstrates the maximum contraction of $M_i$, corresponding to the largest $c_i$, such that ellipsoid of $W_{i+1}^T W_{i+1}$ is still contained in ellipsoid of $c_i M_i$. Similarly, for the case where $c_i < 1$, Fig. 1b demonstrates the minimum extent (the smallest $1/c_i$) to which $M_i$ needs to expand, such that the ellipsoid of $W_{i+1}^T W_{i+1}$ is contained. Algebraically, in both cases, $c_i$ is the ratio of the lengths of the green and pink arrows. By Proposition 2, the resulting ellipsoid (depicted in pink) is $M_i/c_i = W_{i+1}^T W_{i+1} + N/c_i$ for both cases, and is tangent to the ellipsoid of $W_{i+1}^T W_{i+1}$. Moreover, the vector pointing from the origin to the tangency point aligns with the direction of eigenvectors (the grey vector $v$ in the plots) corresponding to the zero eigenvalues of the singular matrix $N \geq 0$.

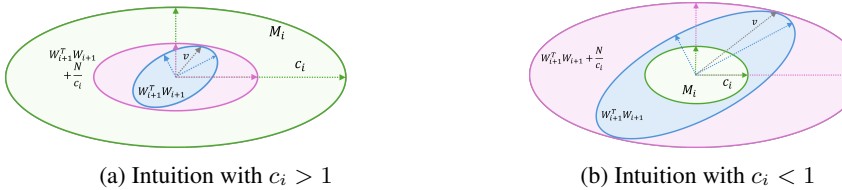

(a) Intuition with $c_i > 1$          (b) Intuition with $c_i < 1$

Figure 1: Geometric Analysis of `ECLipsE`

Combining Proposition 2 and 3, we can derive an optimization problem to sequentially find the appropriate $\Lambda_i$, $i \in \mathbb{Z}_{l-1}$. The first constraint in (12) is quadratic in $\Lambda_i$, which makes it unattractive for practical purposes. Therefore, we apply the Schur Complement to transform it into the linear matrix inequality (LMI) constraint in (8). Thus, the optimization problem in Proposition 3 becomes equivalent to the SDP in (8), yielding algorithm `ECLipsE`. Notice that there are several ways to write the Schur complement of the constraint in (12). We choose this specific structure to avoid singularity of the diagonal entries and ensure positive definiteness.

`ECLipsE-Fast` achieves remarkable speed by further reducing $\Lambda_i$, $i \in \mathbb{Z}_{l-1}$ to a multiple of identity matrix $\lambda_i I$, where $\lambda_i > 0$, and by relaxing the sub-problems. While our goal remains to minimize $\sigma_{max}(\mathcal{F}_{i+1}) = \sigma_{max}\left(W_{i+1}(\Lambda_i - \frac{1}{4}\Lambda_i \mathcal{F}_i \Lambda_i)^{-1} W_{i+1}^T\right)$, we intentionally disregard information from $W_{i+1}$, and instead focus solely on minimizing the spectral norm of $(\Lambda_i - \frac{1}{4}\Lambda_i \mathcal{F}_i \Lambda_i)^{-1}$.

Roughly speaking, a smaller $\sigma_{max}\left((\Lambda_i - \frac{1}{4}\Lambda_i \mathcal{F}_i \Lambda_i)^{-1}\right)$ yields a smaller $\sigma_{max}(\mathcal{F}_{i+1})$. This relaxation allows us to derive a closed-form solution for $\Lambda_i$, $i \in \mathbb{Z}_{l-1}$ as follows.

**Proposition 4.** *Choosing* $\lambda_i = \frac{2}{\sigma_{max}(\mathcal{F}_i)} > 0$ *minimizes* $\sigma_{max}\left((\Lambda_i - \frac{1}{4}\Lambda_i \mathcal{F}_i \Lambda_i)^{-1}\right)$ *where* $\Lambda_i = \lambda_i I$ *under the constraint that* $M_i = \Lambda_i - \frac{1}{4}\Lambda_i \mathcal{F}_i \Lambda_i > 0$. *Moreover, this closed-form solution for* $\lambda_i$ *always satisfies* $M_i > 0$, $i \in \mathbb{Z}_{l-1}$.

By the definition of $\mathcal{F}_i$, Proposition 4 matches with (9), yielding algorithm `ECLipsE-Fast`. Although this relaxation may result in a loss of tightness, the closed-form solution offers the advantage of significantly increased computational speed.

*Geometric Analysis:* We now demonstrate the geometric analysis behind the development of `ECLipsE-Fast` and compare it with `ECLipsE` in the case where $c_i > 1$ (Fig. 2). We also include the case $c_i < 1$ in Appendix B. The key idea behind `ECLipsE-Fast` is that instead of keeping the shape of $M_i$ fixed, and contracting the ellipsoid itself, as in `ECLipsE`, we first find the largest inscribed ball (dark green) for the ellipsoid of $M_i$. Then, we contract this ball to the maximum extent such that it still contains $W_{i+1}^T W_{i+1}$. The resulting ball (dark blue) is precisely the smallest circumscribing ball for the ellipsoid of $W_{i+1}^T W_{i+1}$. Note that this approach serves as an approximation for the process of contraction depicted in Fig. 7b (corresponding to `ECLipsE`), thus yielding a smaller $c_i$. We use this approximation to achieve a closed-form solution, which significantly increases the computational speed.

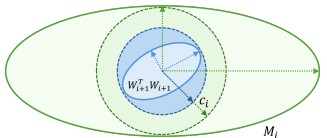 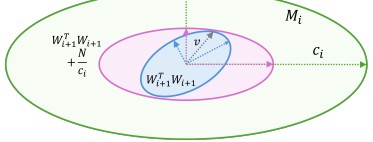

(a) Geometric Intuition of `ECLipsE-Fast`  (b) Geometric Intuition of `ECLipsE`

Figure 2: Comparison between `ECLipsE-Fast` and `ECLipsE` with $c_i > 1$

*Remark* 3. In Lemma 2, the analysis initially fixes the shape of $\mathcal{F}_i$. However, when optimizing $\Lambda_i$, the shape of the feasible region depends on $\mathcal{F}_i$, which can vary with different $\Lambda_{i-1}$, $i \in \mathbb{Z}_l$. Thus, this approximation, which allows for a scalable distributed algorithm to solve the centralized problem (3) introduces an unavoidable but minor tradeoff in achieving global optimality.

## 4 Experiments

We implement our algorithms[†] on randomly generated neural networks and ones trained on the MNIST dataset. The details of the experimental setup, and training of the neural networks (both randomly generated and trained on the MNIST dataset) are described in Appendix D.

**Baselines.**[‡] For `ECLipsE`, $\Lambda_i$, $i \in \mathbb{Z}_{l-1}$ can have different diagonal entries, which benchmarks to LipSDP-Neuron. For `ECLipsE-Fast`, $\Lambda_i = \lambda_i I$, $i \in \mathbb{Z}_{l-1}$, which benchmarks to LipSDP-Layer. Additionally, we compare our Lipschitz estimates to the naive upper bound $L_{naive} = \prod_{i=1}^{l} \|W_i\|_2$[26], CPLip [16] and LipDiff [20]. The codes for these baselines are available at [34, 35, 20]. Note that LipDiff is accelerated using a node with 2 NVIDIA A100 GPUs (80G) and 512 GB of memory.

### 4.1 Randomly Generated Neural Networks

We first consider randomly generated networks, where the number of layers are chosen from $\{2, 5, 10, 20, 30, 50, 75, 100\}$, and number of neurons are chosen from $\{20, 40, 60, 80, 100\}$, amounting to a total of 40 experiments for each algorithm (including the baselines). We quantify the computation time and tightness of the Lipschitz bounds (raw data in Appendix E). The Lipschitz bounds presented in the following figures are normalized to the trivial upper bound for ease of comparison.

---

[†]https://github.com/YuezhuXu/ECLipsE

[‡]Note that SeqLip [13] is also an often-used benchmark; however, we do not consider it since it does not represent a true upper bound for the Lipschitz constant. We also note that we do not include Chordal-LipSDP [15] as a baseline , since only the case where $\tau = 0$ in that work is valid, and all other cases, are no longer valid in certifying the Lipschitz constant as discussed in Remark 1 as well as [15].

**Case 1: Varying network depth (number of layers).** We select a network with 80 neurons per layer, and demonstrate the scalability of our algorithm as network depth increases. Note that all baseline approaches fail to provide a Lipschitz estimate within a computational cutoff time of 15 min for networks larger than this size (see results in Appendix E). As the number of layers increases, the computation time for CPLip algorithm explodes (the algorithm does not return a Lipschitz estimate within the cutoff time beyond 20 layers); however, CPLip provides the most accurate estimates in smaller networks. LipDiff provides inadmissible Lipschitz estimates even for moderate networks, returning as much as 10-100 times the trivial bound (see Table 2a, Appendix E for the estimates). Also, while LipDiff has similar computational time for smaller networks, computational time grows for deeper networks as recorded in Appendix E Table 2b. Consequently, we do not include these results in the plots. LipSDP-Neuron and LipSDP-Layer are also scalable to some extent; however, they fail for a networks of 30 and 50 layers respectively. In contrast, the computation time for `ECLipsE` and `ECLipsE-Fast` stays low and grows only linearly with respect to the number of layers (Fig. 3b). Notably, `ECLipsE-Fast` is significantly faster (thousands of times) than LipSDP-Layer, owing to the closed-form solution at each stage, while `ECLipsE` is also considerably faster than LipSDP-Neuron. The Lipschitz estimates given by algorithms `ECLipsE` and `ECLipsE-Fast` are very close to the ones from LipSDP-Neuron and LipSDP-Layer respectively (Fig. 3a), and outperform the trivial bound. As the number of layers increases, the normalized Lipschitz estimates are smaller, indicating that our algorithms are well-suited to very deep networks.

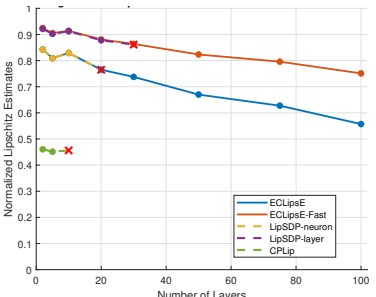
(a) Lipschitz estimates normalized to trivial bound

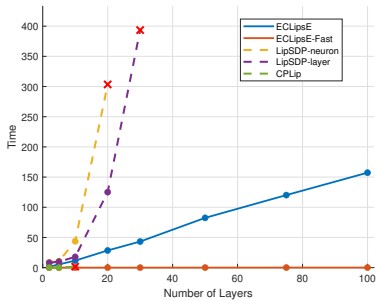
(b) Computation time (seconds)

Figure 3: Performance of `ECLipsE-Fast` and `ECLipsE`, with respect to baselines for increasing network depth, with 80 neurons per layer. The red x markings indicate that the algorithm fails to provide an estimate within the computational cutoff time beyond this network size.

**Case 2: Varying neural network width (number of neurons per layer).** We now examine the performance of our algorithms for wider (more hidden neurons per layer), rather than deeper networks (with more layers), and demonstrate the results for networks with 20 and 50 layers respectively (Fig. 4). While the complete raw data is presented in Appendix E, we discuss the results for 20 and 50 layer networks here, since they represent the network sizes where different baselines fail to return Lipschitz estimates beyond the computation cutoff time of 15 min. Note that while LipDiff also manages to generate estimates for all network sizes in our 50 layers case, it once again provides inadmissible Lipschitz constants, returning as much as $10^4 - 10^6$ times the trivial bound. Therefore, we do not include these results in Fig. 4 (see Tables 3a and 3b in Appendix E for the estimates and computation time.) We can observe from Figs. 4b and 4d that the computation time needed for CPLip, LipSDP-Layer, and LipSDP-Neuron significantly increases with the number of neurons, while the computation time of our method still grows linearly. Meanwhile, the Lipschitz estimates from algorithms `ECLipsE` and `ECLipsE-Fast` are close to the ones from LipSDP-Neuron and LipSDP-Layer respectively (Figs. 4a and 4c). Thus, we can conclude that our method significantly improves scalability for wider neural networks.

**Case 3: Comparison with LipSDP implementations.** In order to address the scalability issue as the size of the network grows, LipSDP utilizes a splitting approach, where the network is split into smaller sub-networks and the Lipschitz constants for each sub-network are composed at the end to obtain the final estimate. We benchmark our approach with respect to the performance of LipSDP-Layer and LipSDP-Neuron considering different sub-network sizes. Note that *our algorithms do not require any splitting*, since they remain scalable to large networks. As the FNNs are larger than the ones in previous cases, we change the cutoff time to 30 minutes. We conduct two sets of experiments

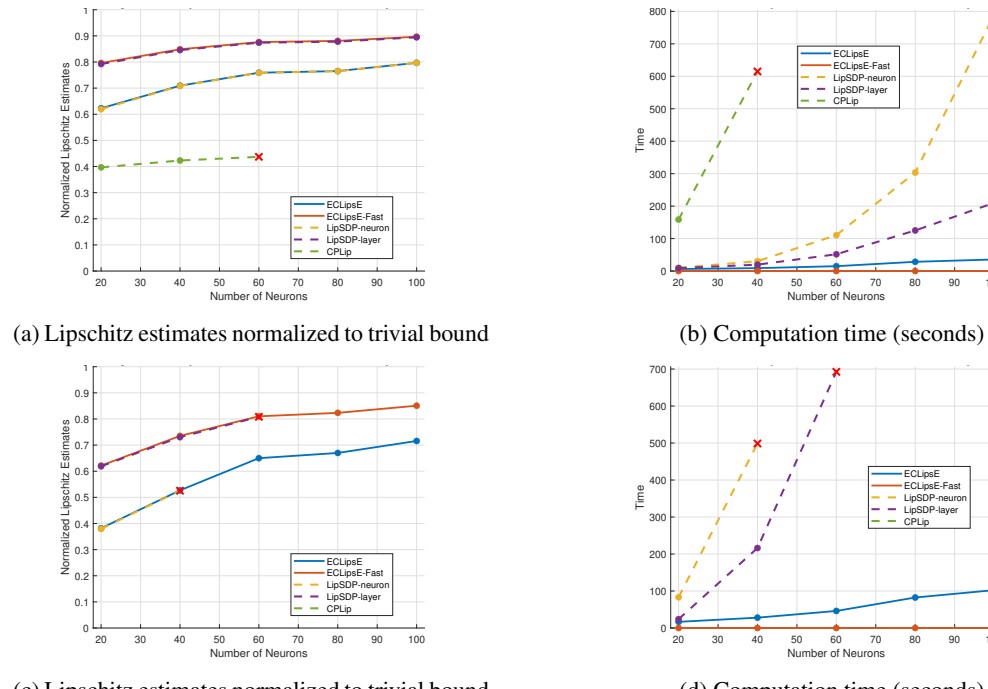

(a) Lipschitz estimates normalized to trivial bound

(b) Computation time (seconds)

(c) Lipschitz estimates normalized to trivial bound

(d) Computation time (seconds)

Figure 4: Performance of **ECLipsE-Fast** and **ECLipsE** with respect to baselines as network width increases, for a randomly generated network with 20 layers ((a) and (b)) and 50 layers ((c) and (d)). The Red x markings indicate that the algorithms fail to provide an estimate within the computational cutoff time of 15 min beyond this network size.

to study how our algorithms perform on considerably deep neural networks and how network width affects these results.

In the first set of experiments, we consider FNNs with 100 layers, with the number of neurons chosen from the set {80,100,120,140,160}. The splitting sizes for LipSDP-Neuron and LipSDP-Layer are 3, 5 and 10. We represent different FNN sizes by shapes and different algorithms by the color in

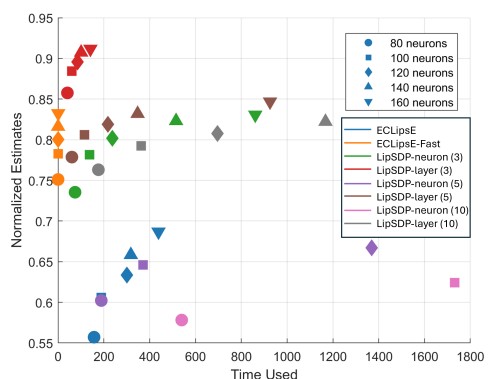

Figure 5: Computation time vs estimation accuracy for **ECLipsE**, **ECLipsE-Fast** and LipSDP splitting with different sub-network sizes.

Fig. 5. By plotting the normalized Lipschitz estimates and computation times on the two axes, we illustrate how efficient and accurate an algorithm is by how close the corresponding data point is to the origin. We observe that all the data points for **ECLipsE-Fast** are at the leftmost extreme of the plot, indicating that it is the most efficient algorithm. Further, **ECLipsE-Fast** also outperforms the red cluster (LipSDP-layer with the network split into 3) in both tightness and speed. Comparing data points of the same shape, **ECLipsE-Fast** outperforms LipSDP-Layer for all sub-network splits both in terms of the Lipschitz estimate and the computation time. Finally, the data points corresponding to **ECLipsE** are clustered at the bottom left, demonstrating that it is relatively more accurate and efficient than all LipSDP methods, no matter how the network is split.

In the second set of experiments, we explore even wider networks. Specifically, we choose a fairly deep neural network with 50 layers and vary the width from 150 to 1000. The splitting size for LipSDP-Neuron and LipSDP-Layer is 5. The resulting Lipschitz estimates (normalized with respect to trivial upper bounds) and the computation time are provided in Tables 4a and 4b of Appendix

E due to space limitations. From these results, we observe that `ECLipsE-Fast` is extremely fast even for very wide networks, with a running time of only 15.63 seconds for a network width of 1000, while the computation time for LipSDP-Layer grows significantly. Also, while `ECLipsE` fails when the width reaches 300, it is comparable to LipSDP-Neuron split into 5 sub-networks in terms of time performance.

***Remark*** 4. We notice that when the neural networks are significantly wide, `ECLipsE` takes more than 30 minutes while `ECLipsE-Fast` remains efficient. This observation can be explained by examining the computational complexity of these algorithms. Note that we directly state the computational complexity of each algorithm here for brevity; the detailed derivations are included in Appendix C. Suppose a neural network has $n$ hidden layers with $m$ neurons. Then, the computational cost for LipSDP and `ECLipsE` are $O(n^4m^4)$ and $O(nm^4)$ respectively. We can observe that the complexity is significantly decreased in terms of the depth, but is the same in terms of the width, immediately indicating the advantage for deep networks. Nevertheless, as $m$ grows, the difference between $O(n^4m^4)$ and $O(nm^4)$ is still drastically enhanced, especially with large $n$. More importantly, for `ECLipsE-Fast`, the computational cost drops to $O(nm^3)$. This is the fastest one can expect if the weights on each layer are treated as a whole.

## 4.2 Neural Networks Trained on MNIST.

We now demonstrate our algorithms on four networks trained on the MNIST dataset (see Appendix D for details) to achieve an accuracy of at least 97%. The resulting networks are not very deep (3 layers), with 100, 200, 300, and 400 neurons. We set a computational cutoff time of 30 min to obtain Lipschitz estimates. As described in the note on Baselines earlier in this section, `ECLipsE` is benchmarked against LipSDP-Neuron and `ECLipsE-Fast` is benchmarked against the faster LipSDP-Layer due to their mathematical structure. From Fig. 6b, we can see that `ECLipsE-Fast` is significantly faster than LipSDP-Layer, while `ECLipsE` is also considerably faster than LipSDP-Neuron. Note that all algorithms provide very similar Lipschitz estimates (Fig. 6a). Therefore, for networks that are not very deep, such as those in this example, `ECLipsE-Fast` is the optimal choice, since it significantly outperforms all algorithms in terms of speed, while the approximation error due to the closed-form solution is not too significant compared to the baselines.

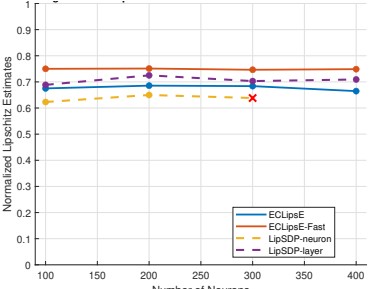
(a) Lipschitz estimates normalized to trivial bound

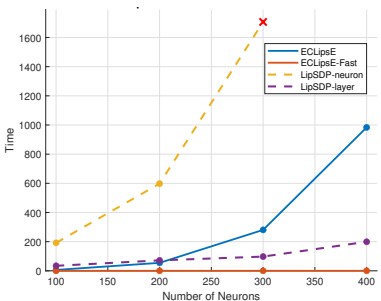
(b) Computation time (seconds)

Figure 6: Performance of `ECLipsE-Fast` and `ECLipsE`, with respect to baselines for increasing number of neurons, for a 3-layer network trained on MNIST. The red x markings indicate that the algorithm fails to provide an estimate within the computational cutoff time beyond this network size.

## 5 Conclusion

We propose a scalable approach to estimate Lipschitz constants for deep neural networks by developing a new matrix decomposition that yields two fast algorithms. Our experiments demonstrate that our algorithms significantly outperform the state-of-the-art in terms of computation speed, while providing comparable Lipschitz estimates. We envision that further computational speedup can be achieved through sparse matrix multiplication and eigenvalue estimation techniques, and leveraging autodiff frameworks, along the lines of [20]. While we can unroll the convolutional layers in CNN structure to a large fully connected neural network layer to apply `ECLipsE` and `ECLipsE-Fast` to estimate Lipschitz constant, better compositional methods that are tailored to feature the convolutional layers are expected for future work. Similarly, other architectures, such as residual networks, present additional challenges due to their unique structures and will be considered in future research.

**Acknowledgment**. This work was partially supported by the Air Force Office of Scientific Research grant, FA9550-23-1-0492.

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

# Appendices

## A   Technical Proofs

**Proof of Relation in Assumption 1.** We show how the slope-restrictedness of the activation function implies (2). The inequality $\alpha(v_1 - v_2) \leq \phi(v_1) - \phi(v_2) \leq \beta(v_1 - v_2)$ that holds elementwise yields

$$\lambda^i \left[(\phi(v_1) - \phi(v_2)) - \alpha(v_1 - v_2)\right]_i \times \left[(\phi(v_1) - \phi(v_2)) - \beta(v_1 - v_2)\right]_i \leq 0, \ \forall i \in \mathbb{R}^n, \forall \lambda^i \geq 0,$$

where subscript $i$ indexes the $i^{th}$ element of the vector.
Summing all the inequalities for $i \in \mathbb{R}^n$ and letting $\Lambda = diag(\lambda^1, \lambda^2, ..., \lambda^n)$, we have

$$\left[(\phi(v_1) - \phi(v_2)) - \alpha(v_1 - v_2)\right]^T \Lambda \left[(\phi(v_1) - \phi(v_2)) - \beta(v_1 - v_2)\right] \leq 0.$$

In other words, we have the following quadratic inequality

$$(\phi(v_1) - \phi(v_2))^T \Lambda (\phi(v_1) - \phi(v_2)) - \frac{\alpha + \beta}{2}(\phi(v_1) - \phi(v_2))^T \Lambda (v_1 - v_2)$$

$$-\frac{\alpha + \beta}{2}(v_1 - v_2)^T \Lambda (\phi(v_1) - \phi(v_2)) + \alpha\beta(v_1 - v_2)^T \Lambda (v_1 - v_2) \leq 0,$$

which can be directly rewritten as (2) with $p = \alpha\beta$ and $m = \frac{\alpha+\beta}{2}$.

**Proof of Theorem 1**. For any two inputs $z_1^{(0)}$ and $z_2^{(0)}$, let $z_1^{(i)}, z_2^{(i)}, v_1^{(i)}, v_2^{(i)}, i \in \mathbb{Z}_l$ be computed as in (1). Define $\Delta z^{(i)} = z_1^{(i)} - z_2^{(i)}, i \in \{0\} \cup \mathbb{Z}_l$ and $\Delta v^{(j)} = v_1^{(j)} - v_2^{(j)}, j \in \mathbb{Z}_l$. By both left and right multiplying the left hand side in (3) by the vector $[\Delta z^{(0)}, \Delta z^{(1)}, ..., \Delta z^{(l-1)}]^T$, we have

$$(\Delta z^{(0)})^T \Delta z^{(0)} + \sum_{i=1}^{l-1} \left( \begin{bmatrix} \Delta z^{(i-1)} \\ \Delta z^{(i)} \end{bmatrix}^T \begin{bmatrix} pW_i^T \Lambda_i W_i & -mW_i^T \Lambda_i \\ -m\Lambda_i W_i & \Lambda_i \end{bmatrix} \begin{bmatrix} \Delta z^{(i-1)} \\ \Delta z^{(i)} \end{bmatrix} \right) \tag{13}$$

$$- F(\Delta z^{(l-1)}) W_l^T W_l \Delta z^{(l-1)} > 0.$$

Now, we show that every summand in the above inequality (13) is negative semidefinite. In fact, with Assumption 1, using notation $\Delta v^{(i)} = v_1^{(i)} - v_2^{(i)}$ and $\Delta\phi(v^{(i)}) = \phi(v_1^{(i)}) - \phi(v_2^{(i)}), i \in \mathbb{Z}_l$, and taking $\Lambda = \Lambda_i$, we can write

$$\begin{bmatrix} \Delta v^{(i)} \\ \Delta\phi(v^{(i)}) \end{bmatrix}^T \begin{bmatrix} p\Lambda_i & -m\Lambda_i \\ -m\Lambda_i & \Lambda_i \end{bmatrix} \begin{bmatrix} \Delta v^{(i)} \\ \Delta\phi(v^{(i)}) \end{bmatrix} \leq 0. \tag{14}$$

Note that $\Delta v^{(i)} = (W_i z_1^{(i-1)} + b_i) - (W_i z_2^{(i-1)} + b_i) = W_i \Delta z^{(i-1)}$ and $\Delta\phi(v^i) = \Delta z^i$. We can express these relationships in matrix form as

$$\begin{bmatrix} \Delta v^{(i)} \\ \Delta\phi(v^{(i)}) \end{bmatrix} = \begin{bmatrix} W_i & 0 \\ 0 & I \end{bmatrix} \begin{bmatrix} \Delta z^{(i-1)} \\ \Delta z^{(i)} \end{bmatrix}. \tag{15}$$

Substituting (15) into (14), we have

$$\begin{bmatrix} \Delta z^{(i-1)} \\ \Delta z^{(i)} \end{bmatrix}^T \begin{bmatrix} pW_i^T \Lambda_i W_i & -mW_i^T \Lambda_i \\ -m\Lambda_i W_i & \Lambda_i \end{bmatrix} \begin{bmatrix} \Delta z^{(i-1)} \\ \Delta z^{(i)} \end{bmatrix} \leq 0. \tag{16}$$

Combining (13) and (16), we have

$$(\Delta z^{(0)})^T \Delta z^{(0)} - F(\Delta z^{(l-1)}) W_l^T W_l \Delta z^{(l-1)} \geq 0. \tag{17}$$

Similarly, as the last layer is a linear layer, $\Delta z^{(l)} = \Delta v^{(l)} = W_l \Delta z^{(l-1)}$. Then (17) is exactly

$$(\Delta z^l)^T \Delta z^l \leq \frac{1}{F}(\Delta z^{(0)})^T \Delta z^{(0)},$$

yielding a upper bound $\sqrt{1/F}$ for the Lipschitz constant.

**Proof of Theorem 2**. Applying Lemma 2 in [31], we define

$$\mathcal{R}_i = -\frac{1}{2} W_{i-1}^T \Lambda_{i-1}, i \in \mathbb{Z}_l, \tag{18}$$

and

$$\mathcal{P}_i = \begin{cases} I & \text{if } i = 0, \\ \Lambda_i & \text{if } i \in \mathbb{Z}_{l-2}, \\ \Lambda_i - F W_l^T W_l & \text{if } i = l-1. \end{cases} \tag{19}$$

Then, with $\mathcal{P}_i$ and $\mathcal{R}_i$ defined above, we directly have

$$M_i = \begin{cases} X_i & \text{if } i \in \{0\} \bigcup \mathbb{Z}_{l-2}, \\ X_i + F W_l^T W_l & \text{if } i = l-1. \end{cases} \tag{20}$$

In other words, the sufficient and necessary condition $X_i > 0, \forall i \in \{0\} \cup \mathbb{Z}_{l-1}$ is equivalent to

$$M_i > 0, \quad \forall i \in \mathbb{Z}_{l-2}, \qquad M_{l-1} - F W_l^T W_l > 0, \tag{21}$$

which is the same as (21).

**Proof of Proposition 1**. As $M_i > 0$, $i \in \mathbb{Z}_{l-2}$ has been guaranteed, it remains to ensure that $M_{l-1} - F W_l^T W_l > 0$ by Theorem 3. This is equivalent to $M_{l-1}/F > W_l^T W_l$. Therefore, the smallest possible $1/F$ is $\sigma_{max}(W_l^T W_l (M_{l-1})^{-1})$. Then by Theorem 1, the upper bound for the Lipschitz constant is $\sqrt{1/F} = \sqrt{\sigma_{max}(W_l^T W_l (M_{l-1})^{-1})}$.

**Proof of Lemma 1**. $M_i > 0$ indicates $(M_i)^{-1} > 0$. Then for $\forall v_0 \neq 0$,

$$v_0^T W_{i+1} (M_i)^{-1} W_{i+1}^T v_0 = (W_{i+1}^T v_0)^T (M_i)^{-1} W_{i+1}^T v_0 \geq 0, \tag{22}$$

meaning that all eigenvalues of $W_{i+1}(M_i)^{-1} W_{i+1}^T$ are non-negative. As $W_{i+1} \neq 0$, we know that the largest eigenvalue is positive and should be the same as $\sigma_{max}\left(W_{i+1}(M_i)^{-1} W_{i+1}^T\right) > 0$.
Now consider any non-zero eigenvalue $\lambda_a$ of matrix $W_{i+1}(M_i)^{-1} W_{i+1}^T$ and let $v_a \neq 0$ be its corresponding eigenvector. Then,

$$W_{i+1}(M_i)^{-1} W_{i+1}^T v_a = \lambda_a v_a. \tag{23}$$

Left multiplying both sides with $W_{i+1}^T$, we have

$$W_{i+1}^T W_{i+1} (M_i)^{-1} (W_{i+1}^T v_a) = W_{i+1}^T W_{i+1}(M_i)^{-1} W_{i+1}^T v_a = W_{i+1}^T \lambda_a v_a = \lambda_a (W_{i+1}^T v_a). \tag{24}$$

As $\lambda_a \neq 0$, we know that $W_{i+1}^T v_a \neq 0$ from (23). (Otherwise, $\lambda_a v_a = 0$ with $v_a \neq 0$ will lead to $\lambda_a = 0$). With $W_{i+1}^T v_a \neq 0$, (24) implies that $\lambda_a$ is also an eigenvalue of $W_{i+1}^T W_{i+1}(M_i)^{-1}$ corresponding to eigenvector $W_{i+1}^T v_a \neq 0$.
Conversely, for any non-zero eigenvalue $\lambda_b$ of $W_{i+1}^T W_{i+1}(M_i)^{-1}$ corresponding to eigenvector $v_b \neq 0$, we have

$$W_{i+1}^T W_{i+1}(M_i)^{-1} v_b = \lambda_b v_b. \tag{25}$$

Let $v_c = \frac{1}{\lambda_b} W_{i+1}(M_i)^{-1} v_b$. We have $v_b = W_{i+1}^T v_c$. Then we substitute $v_b$ on the both sides in (25) and obtain

$$W_{i+1}^T W_{i+1}(M_i)^{-1} W_{i+1}^T v_c = \lambda_b W_{i+1}^T v_c.$$

Left multiplying both side with $W_{i+1}(M_i)^{-1}$, we further have

$$W_{i+1}(M_i)^{-1} W_{i+1}^T W_{i+1}(M_i)^{-1} W_{i+1}^T v_c = \lambda_b W_{i+1}(M_i)^{-1} W_{i+1}^T v_c.$$

Let $v_d = W_{i+1}(M_i)^{-1} W_{i+1}^T v_c$, we finally have

$$W_{i+1}(M_i)^{-1} W_{i+1}^T v_d = \lambda_b v_d.$$

Similarly, we can conclude that $\lambda_b \neq 0$ is the eigenvalue of $W_{i+1}(M_i)^{-1} W_{i+1}^T$ and $v_d$ is the corresponding eigenvector.

**Proof of Lemma 2.** To prove that $\{\Lambda_i : M_i > 0, \Lambda_i \in \mathbb{D}_+\} \subseteq \{\Lambda_i : \tilde{M}_i > 0, \Lambda_i \in \mathbb{D}_+\}$, it suffices to show that $\tilde{M}_i - M_i \geq 0$ for any $\gamma \in (0, 1)$. In fact,

$$\tilde{M}_i - M_i = \Lambda_i - \frac{1}{4}\Lambda_i \mathcal{F}_i \Lambda_i - (\Lambda_i - \frac{\gamma}{4}\Lambda_i \mathcal{F}_i \Lambda_i) = \frac{1-\gamma}{4}\Lambda_i \mathcal{F}_i \Lambda_i. \tag{26}$$

At stage $i$, $M_{i-1} > 0$ is guaranteed. Then, similar to (22), we know that $\mathcal{F}_i = W_i(M_{i-1})^{-1}W_i^T \geq 0$ Since $\Lambda_i \in \mathbb{D}_+$ and $0 < \gamma < 1$, implying $\frac{1-\gamma}{4} > 0$, we have $\frac{1-\gamma}{4}\Lambda_i \mathcal{F}_i \Lambda_i \geq 0$.

**Proof of Proposition 2.** Recall that by definition, $\mathcal{F}_{i+1} = W_{i+1}(M_i)^{-1}W_{i+1}^T$. Applying Lemma 1, we have $\sigma_{max}(\mathcal{F}_{i+1}) = \sigma_{max}(W_{i+1}^T W_{i+1}(M_i)^{-1})$. Meanwhile, with $M_i = c_i W_{i+1}^T W_{i+1} + N$,

$$W_{i+1}^T W_{i+1}(M_i)^{-1} = (W_{i+1}^T W_{i+1} + \frac{N}{c_i})(c_i W_{i+1}^T W_{i+1} + N)^{-1} - \frac{N}{c_i}(c_i W_{i+1}^T W_{i+1} + N)^{-1}$$

$$= \frac{1}{c_i}I - \frac{N}{c_i}(M_i)^{-1}. \tag{27}$$

WIth $N \geq 0$ and $M_i > 0$ after $\Lambda_i$ is decided, we show that $\frac{N}{c_i}(M_i)^{-1}$ only has non-negative eigenvalues. As $M_i > 0$ is guaranteed to symmetric according to (7), there exists a symmetric square root for $(M_i)^{-1}$ and we denote it to be $(M_i)^{-\frac{1}{2}}$. Then $N(M_i)^{-1}$ is similar to $(M_i)^{-\frac{1}{2}}N(M_i)^{-\frac{1}{2}}$, thus sharing the same eigenvalues. Furthermore, for $\forall x \neq 0$,

$$x^T (M_i)^{-\frac{1}{2}}N(M_i)^{-\frac{1}{2}}x = \left((M_i)^{-\frac{1}{2}}x\right)^T N \left((M_i)^{-\frac{1}{2}}x\right) \geq 0.$$

It indicates $(M_i)^{-\frac{1}{2}}N(M_i)^{-\frac{1}{2}}$ only has non-negative eigenvalues. The first equation holds by the symmetry of $(M_i)^{-\frac{1}{2}}$ and the second inequality is because of the definition of positive semi-definiteness of $N$. Therefore, with $c_i > 0$, the eigenvalues of $\frac{N}{c_i}(M_i)^{-1} = \frac{1}{c_i}N(M_i)^{-1}$ are all non-negative. and all eigenvalues of $W_{i+1}^T W_{i+1}(M_i)^{-1}$ should be less than or equal to $1/c_i$. On the other hand, as $N$ is singular, $N(M_i)^{-1}$ is also singular, thus having eigenvalue 0. Therefore,

$$\sigma_{max}(\mathcal{F}_{i+1}) = \sigma_{max}(W_{i+1}^T W_{i+1}(M_i)^{-1}) = 1/c_i.$$

**Proof of Proposition 3.** We first show that if $M_{i-1} > 0$, the feasible region is always non-empty for $\forall i \in \mathcal{Z}_{l-1}$. We use $\sigma$ to denote $\sigma_{max}\left(W_i(M_{i-1})^{-1}W_i^T\right) = \sigma_{max}(\mathcal{F}_i)$. We take $\Lambda_i = \frac{2}{\sigma}I$ and $c_i = \frac{0.9}{\sigma \cdot \sigma_{max}\left(W_{i+1}^T W_{i+1}\right)}$. As $W_i \neq 0$, $i \in \mathbb{Z}_l$, we have $\sigma > 0$ and $\sigma_{max}(\mathcal{F}_i) > 0$, ensuring $\Lambda_i \in \mathbb{D}_+$ and $c_i > 0$. Further,

$$\Lambda_i - \frac{1}{4}\Lambda_i W_i(M_{i-1})^{-1}W_i^T \Lambda_i - c_i(W_{i+1}^T W_{i+1})$$

$$\geq \frac{2}{\sigma}I - \frac{1}{4}\frac{4}{\sigma^2}\sigma I - \frac{0.9}{\sigma \cdot \sigma_{max}\left(W_{i+1}^T W_{i+1}\right)}(W_{i+1}^T W_{i+1}) \tag{28}$$

$$> \frac{1}{\sigma}I - \frac{0.9}{\sigma}I > 0.$$

Therefore, the feasible region at least includes the $\Lambda_i$ and $c_i$ we specified, and is thus not empty. To make the feasibility complete, we prove that $M_i > 0$, $\forall i \in \{0\}\bigcup \mathcal{Z}_{l-1}$ by induction. When $i = 0$, $M_i = M_0 = I > 0$. When it comes to stage $i$, we have by induction that $M_{i-1} > 0$ is true. As $\Lambda_i$ are obtained satisfying (12) with $c_i > 0$ and recall the recursive relation for $M_i$, $i \in \mathbb{Z}_{l-1}$ to be $M_i = \Lambda_i - \frac{1}{4}\Lambda_i W_i(M_{i-1})^{-1}W_i^T \Lambda_i$, we have $M_i > c_i W_{i+1}^T W_{i+1} \geq 0$.

We now prove by contradiction that the optimal value $c_i$ is the largest constant such that $M_i$ can be written as $M_i = c_i W_{i+1}^T W_{i+1} + N$, where $N$ is some singular matrix that $N \geq 0$. $M_i = c_i W_{i+1}^T W_{i+1} + N$. Suppose there exists a $\hat{c}_i > c_i$ such that it satisfies all the constraints. Then, from the first constraint in (12), we have

$$(c_i - \hat{c}_i)W_{i+1}^T W_{i+1} + N = M_i - \hat{c}_i(W_{i+1}^T W_{i+1}) > 0 \tag{29}$$

Let $v \neq 0$ be the eigenvector of $N$ corresponding to eigenvalue 0. Observe that

$$v^T \left((c_i - \hat{c}_i)W_{i+1}^T W_{i+1} + N\right)v$$

$$= v^T \left((c_i - \hat{c}_i)W_{i+1}^T W_{i+1}\right)v + 0 \tag{30}$$

$$= (c_i - \hat{c}_i)v^T \left(W_{i+1}^T W_{i+1}\right)v \leq 0.$$

The last step holds because $c_i - \hat{c}_i < 0$ by definition of $\hat{c}_i$. It contradicts (29).

**Proof of Proposition 4.** When $\Lambda_i = \lambda_i I$,

$$\left( \Lambda_i - \frac{1}{4} \Lambda_i \mathcal{F}_i \Lambda_i \right)^{-1} = \left( \lambda_i I - \frac{1}{4} \lambda_i^2 \mathcal{F}_i \right)^{-1}.$$

As $M_i = \Lambda_i - \frac{1}{4} \Lambda_i \mathcal{F}_i \Lambda_i > 0$, we have

$$\sigma_{max} \left( \left( \lambda_i I - \frac{1}{4} \lambda_i^2 \mathcal{F}_i \right)^{-1} \right) = \frac{1}{\lambda_i - \lambda_i^2 \sigma_{max}(\mathcal{F}_i)/4} \tag{31}$$

Minimizing the above spectrum is equivalent to maximizing the denominator $\lambda_i - \lambda_i^2 \sigma_{max}(\mathcal{F}_i)/4$ in (31), which is quadratic in $\lambda_i$. To find the optimal $\lambda_i$, we set the derivative of the denominator with respect to $\lambda_i$ to be 0, and obtain the closed-form solution $\lambda_i = \frac{2}{\sigma_{max}(\mathcal{F}_i)}$.

Moreover, with $\Lambda_i = \frac{2}{\sigma_{max}(\mathcal{F}_i)} I$ and $\mathcal{F}_i > 0$ guaranteed at stage $i$, we have

$$M_i = \Lambda_i - \frac{1}{4} \Lambda_i \mathcal{F}_i \Lambda_i = \frac{1}{\sigma_{max}(\mathcal{F}_i)} I > 0.$$

## B  Geometric Analysis for `ECLipsE-Fast`

The geometric analysis for algorithm `ECLipsE-Fast` analogous to Fig. 2, comparing the case where $c_i > 1$ and in other cases are shown in Fig. 7.

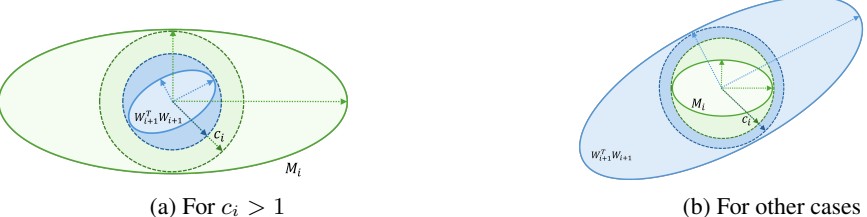

(a) For $c_i > 1$            (b) For other cases

Figure 7: Geometric Intuition of `ECLipsE-Fast` with $c_i > 1$ and otherwise.

## C  Computational Complexity Derivation

We derive the time complexity for both `ECLipsE` and `ECLipsE-Fast` in detail here. Suppose a neural network has $n$ hidden layers with $m$ neurons. Then, the large matrix in Theorem 1 has dimension $nm + O(1)$ and the decision variable is of size $nm + O(1)$. Note that the computational complexity for solving an LMI with the size of the matrix constraint being size $A$ and the number of decision variables being $B$ is $O(A^3 + A^2 B^2)$. Therefore, the computational cost for LipSDP is $O((nm + O(1))^3 + (nm + O(1))^2 (nm + O(1))^2) = O(n^4 m^4)$. Contrarily, `ECLipsE` solves $n$ sub-problems as in Eq. (8), each involving a matrix of size $O(m)$ and $m$ decision variables. The corresponding total computational cost is $n \times (O(m^3 + m^2 m^2)) = O(nm^4)$. This directly indicates the advantage of `ECLipsE` for deep networks. Also, as $m$ grows, the difference between $O(n^4 m^4)$ and $O(nm^4)$ is still significantly enhanced, especially with large $n$. Regarding `ECLipsE-Fast`, we note that we do not need to solve any SDPs and the computational cost drops to $n \times O(m^3) = O(nm^3)$. This is the fastest one can expect if the weights on each layer are treated as a whole.

## D  Experimental Setup and Data Generation

**Experimental Setup.** All experiments are implemented on a Windows laptop with a 12-core CPU with 16GB of RAM.

**Randomly Generated Neural Networks.** We set the input dimension to be 4 and the output dimension to be 1. The activation functionsare chosen to be ReLU, and the number of neurons in each hidden layer is set to be the same. We randomly generate weights for each layer to follow the normal

distribution. Also, in order to avoid the case where the Lipschitz constant is too large or too small and may cause numerical issues, we scale the weights on each layer such that the is norm randomly chosen in $[0.4, 1.8]$, following a uniform distribution.

**MNIST.** For training on the dataset MNIST, the input dimension is 784 and output dimension is 10, which is compatible with the dataset. he activation functionsare chosen to be ReLU, and the number of neurons in each hidden layer is set to be the same. We train neural networks using the SGD optimizer with a learning rate of 0.01 and momentum of 0.9 until they achieve at least 97% accuracy on test data.

# E    Additional Experimental Results

The Lipschitz constant estimates and computation times for randomly generated neural networks with the number of layers chosen from $\{2, 5, 10, 20, 30, 50, 75, 100\}$, and number of neurons are chosen from $\{20, 40, 60, 80, 100\}$, are provided below.

| | Table 1a: Lipschitz constant estimates | | | | | | | |
|---|---|---|---|---|---|---|---|---|
| | Neurons\Layers | 2 | 5 | 10 | 20 | 30 | 50 | 75 | 100 |

| Trivial Bound | Neurons\Layers | 2 | 5 | 10 | 20 | 30 | 50 | 75 | 100 |
|---|---|---|---|---|---|---|---|---|---|
| | 20 | 1.020 | 3.171 | 1.173 | 6.725 | 3.430 | 49.696 | 1.091 | 0.002 |
| | 40 | 0.952 | 3.356 | 0.910 | 3.431 | 0.004 | 260.807 | 2.895 | 0.119 |
| | 60 | 1.433 | 2.830 | 0.040 | 0.067 | 0.706 | 0.013 | 16.433 | 8.890 |
| | 80 | 0.875 | 0.418 | 0.681 | 4.023 | 0.010 | 0.057 | 1.291 | 0.054 |
| | 100 | 1.046 | 0.626 | 4.144 | 0.346 | 2.521 | 46.466 | 6.933 | 95.263 |
| ECLipsE | Neurons\Layers | 2 | 5 | 10 | 20 | 30 | 50 | 75 | 100 |
| | 20 | 0.856 | 2.485 | 0.822 | 4.189 | 1.985 | 18.974 | 0.290 | 0.000 |
| | 40 | 0.775 | 2.696 | 0.722 | 2.434 | 0.003 | 137.421 | 1.413 | 0.043 |
| | 60 | 1.207 | 2.391 | 0.031 | 0.051 | 0.480 | 0.008 | 9.187 | 4.388 |
| | 80 | 0.737 | 0.338 | 0.565 | 3.078 | 0.007 | 0.039 | 0.810 | 0.030 |
| | 100 | 0.884 | 0.527 | 3.414 | 0.276 | 1.904 | 33.261 | 4.524 | 57.734 |
| ECLipsE-Fast | Neurons\Layers | 2 | 5 | 10 | 20 | 30 | 50 | 75 | 100 |
| | 20 | 0.941 | 2.825 | 0.990 | 5.354 | 2.612 | 30.884 | 0.568 | 0.001 |
| | 40 | 0.868 | 3.030 | 0.814 | 2.912 | 0.003 | 191.736 | 2.026 | 0.072 |
| | 60 | 1.324 | 2.611 | 0.035 | 0.059 | 0.588 | 0.010 | 12.355 | 6.285 |
| | 80 | 0.809 | 0.378 | 0.622 | 3.544 | 0.009 | 0.047 | 1.027 | 0.041 |
| | 100 | 0.968 | 0.577 | 3.779 | 0.310 | 2.204 | 39.529 | 5.633 | 74.570 |
| LipSDP-Neuron | Neurons\Layers | 2 | 5 | 10 | 20 | 30 | 50 | 75 | 100 |
| | 20 | 0.856 | 2.481 | 0.819 | 4.165 | 1.978 | 18.851 | 0.287 | |
| | 40 | 0.775 | 2.693 | 0.721 | 2.430 | 0.003 | 137.025 | | |
| | 60 | 1.207 | 2.390 | 0.031 | 0.051 | 0.479 | | | |
| | 80 | 0.737 | 0.338 | 0.564 | 3.077 | | | | |
| | 100 | 0.884 | 0.526 | 3.413 | 0.276 | | | | |
| LipSDP-Layer | Neurons\Layers | 2 | 5 | 10 | 20 | 30 | 50 | 75 | 100 |
| | 20 | 0.938 | 2.814 | 0.985 | 5.327 | 2.607 | 30.763 | 0.565 | |
| | 40 | 0.863 | 3.019 | 0.812 | 2.901 | 0.003 | 190.489 | | |
| | 60 | 1.319 | 2.606 | 0.035 | 0.059 | 0.584 | 0.010 | | |
| | 80 | 0.806 | 0.377 | 0.621 | 3.531 | 0.009 | | | |
| | 100 | 0.965 | 0.575 | 3.770 | 0.310 | 2.197 | | | |
| CPLip | Neurons\Layers | 2 | 5 | 10 | 20 | 30 | 50 | 75 | 100 |
| | 20 | 0.469 | 1.408 | 0.493 | 2.669 | | | | |
| | 40 | 0.432 | 1.510 | 0.406 | 1.451 | | | | |
| | 60 | 0.660 | 1.303 | 0.018 | 0.029 | | | | |
| | 80 | 0.403 | 0.189 | 0.311 | | | | | |
| | 100 | 0.483 | 0.288 | 1.885 | | | | | |

| Table 1b: Computation time (seconds) | | | | | | | | |
|---|---|---|---|---|---|---|---|---|
| Trivial Bound | Neurons\Layers | 2 | 5 | 10 | 20 | 30 | 50 | 75 | 100 |
| | N/A | | | | | | | | |
| ECLipsE | Neurons\Layers | 2 | 5 | 10 | 20 | 30 | 50 | 75 | 100 |
| | 20 | 0.374 | 1.393 | 2.776 | 6.243 | 10.246 | 16.731 | 24.533 | 37.118 |
| | 40 | 0.572 | 2.115 | 4.263 | 8.944 | 15.730 | 27.977 | 36.769 | 52.201 |
| | 60 | 1.007 | 3.551 | 7.768 | 14.938 | 25.616 | 46.128 | 72.479 | 109.913 |
| | 80 | 1.381 | 5.428 | 11.650 | 28.458 | 43.255 | 82.461 | 120.167 | 157.274 |
| | 100 | 2.346 | 7.818 | 18.265 | 35.685 | 53.392 | 102.400 | 160.400 | 188.536 |
| ECLipsE-Fast | Neurons\Layers | 2 | 5 | 10 | 20 | 30 | 50 | 75 | 100 |
| | 20 | 0.001 | 0.002 | 0.001 | 0.002 | 0.003 | 0.006 | 0.007 | 0.010 |
| | 40 | 0.002 | 0.004 | 0.008 | 0.018 | 0.029 | 0.036 | 0.060 | 0.057 |
| | 60 | 0.002 | 0.005 | 0.010 | 0.026 | 0.038 | 0.057 | 0.076 | 0.083 |
| | 80 | 0.004 | 0.009 | 0.024 | 0.051 | 0.056 | 0.089 | 0.127 | 0.136 |
| | 100 | 0.007 | 0.016 | 0.021 | 0.070 | 0.058 | 0.095 | 0.151 | 0.190 |
| LipSDP-Neuron | Neurons\Layers | 2 | 5 | 10 | 20 | 30 | 50 | 75 | 100 |
| | 20 | 5.684 | 6.691 | 8.944 | 9.876 | 15.356 | 83.294 | 153.800 | 373.500 |
| | 40 | 6.974 | 8.192 | 12.519 | 30.342 | 87.703 | 498.750 | >15min | |
| | 60 | 8.285 | 9.410 | 18.654 | 110.670 | 438.040 | >15min | | |
| | 80 | 8.812 | 10.749 | 43.734 | 303.440 | >15min | | | |
| | 100 | 8.876 | 15.009 | 88.894 | 789.330 | | | | |
| LipSDP-Layer | Neurons\Layers | 2 | 5 | 10 | 20 | 30 | 50 | 75 | 100 |
| | 20 | 5.594 | 5.941 | 7.800 | 9.013 | 9.831 | 23.968 | 117.23 | 342.93 |
| | 40 | 6.941 | 7.616 | 8.829 | 19.676 | 40.463 | 216.22 | >15min | |
| | 60 | 7.849 | 8.790 | 12.591 | 51.714 | 140.270 | 692.47 | | |
| | 80 | 8.087 | 9.834 | 17.815 | 125.050 | 393.480 | >15min | | |
| | 100 | 8.431 | 10.356 | 33.859 | 210.090 | 687.710 | | | |
| CPLip | Neurons\Layers | 2 | 5 | 10 | 20 | 30 | 50 | 75 | 100 |
| | 20 | ≈0 | 0.001 | 0.105 | 158.603 | >15min | | | |
| | 40 | ≈0 | 0.003 | 0.385 | 614.917 | | | | |
| | 60 | ≈0 | 0.006 | 0.59 | >15min | | | | |
| | 80 | ≈0 | 0.018 | 1.633 | >15min | | | | |
| | 100 | ≈0 | 0.079 | 3.851 | | | | | |

| Table 2a: Normalized Lipschitz Estimates for Randomly Generated NN with 80 Neurons | | | | | | |
|---|---|---|---|---|---|---|
| Layers | ECLipsE | ECLipsE-Fast | LipDiff | LipSDP-Neuron | LipSDP-Layer | CP-Lip |
| 20 | 0.765184 | 0.88079 | 1.72459 | 0.76481 | 0.877786 | >15min |
| 30 | 0.737564 | 0.863682 | 155.8985 | >15min | 0.861134 | |
| 50 | 0.669903 | 0.823353 | 5.320799 | | >15min | |
| 75 | 0.627432 | 0.795877 | 18.57997 | | | |
| 100 | 0.557117 | 0.751101 | >15min | | | |

| Table 2b: Time used (sec) for Randomly Generated NN with 80 Neurons | | | | | | |
|---|---|---|---|---|---|---|
| Layers | ECLipsE | ECLipsE-Fast | LipDiff | LipSDP-Neuron | LipSDP-Layer | CP-Lip |
| 20 | 28.45839 | 0.0515 | 22.23 | 303.44 | 125.05 | >15min |
| 30 | 43.25548 | 0.05645 | 51.22 | >15min | 393.48 | |
| 50 | 82.46052 | 0.089058 | 178.03 | | >15min | |
| 75 | 120.1665 | 0.126933 | 532 | | | |
| 100 | 157.2741 | 0.136244 | >15min | | | |

| Table 3a: Normalized Lipschitz Estimates for Randomly Generated NN with 50 layers | | | | | | |
|---|---|---|---|---|---|---|
| Neurons | ECLipsE | ECLipsE-Fast | LipDiff | LipSDP-Neuron | LipSDP-Layer | CP-Lip |
| 20 | 0.381796 | 0.621443 | 118628.2 | 0.379323 | 0.619014 | >15min |
| 40 | 0.526908 | 0.735163 | 2635012.67 | 0.525388 | 0.730384 | |
| 60 | 0.649970 | 0.810128 | 23.46069 | >15min | 0.808237 | |
| 80 | 0.669903 | 0.823353 | 5.505175 | | >15min | |
| 100 | 0.71581 | 0.850702 | 10622.75 | | | |

| Table 3b: Time used (sec) for Randomly Generated NN with 50 Layers | | | | | | |
|---|---|---|---|---|---|---|
| Neurons | ECLipsE | ECLipsE-Fast | LipDiff | LipSDP-Neuron | LipSDP-Layer | CP-Lip |
| 20 | 16.73062 | 0.005613 | 12.31672 | 83.294 | 23.968 | >15min |
| 40 | 27.97682 | 0.03643 | 34.29124 | 498.75 | 216.22 | |
| 60 | 46.12812 | 0.056791 | 86.27217 | >15min | 692.47 | |
| 80 | 82.46052 | 0.089058 | 178.2235 | | >15min | |
| 100 | 102.4001 | 0.095034 | 327.946 | | | |

| Table 4a: Normalized Lipschitz Estimates for Randomly Generated NN with 50 Layers | | | | |
|---|---|---|---|---|
| Neurons | ECLipsE | ECLipsE-Fast | LipSDP-Neuron Split by 5 | LipSDP-Layer Split by 5 |
| 150 | 0.743745 | 0.867548 | 0.758217 | 0.87342 |
| 200 | 0.773494 | 0.883758 | 0.785171 | 0.888306 |
| 300 | >30min | 0.897008 | >30min | 0.899164 |
| 400 | | 0.899916 | | >30min |
| 500 | | 0.903529 | | |
| 1000 | | 0.912093 | | |

| Table 4b: Time Used (sec) for Randomly Generated NN with 50 Layers | | | | |
|---|---|---|---|---|
| Neurons | ECLipsE | ECLipsE-Fast | LipSDP-Neuron Split by 5 | LipSDP-Layer Split by 5 |
| 150 | 387.7 | 0.387262 | 451.07 | 93.129 |
| 200 | 1386.6 | 0.584115 | 1377.9 | 210.16 |
| 300 | >30min | 1.321177 | >30min | 612.47 |
| 400 | | 2.657505 | | 2110.9 |
| 500 | | 3.7435 | | >30min |
| 1000 | | 15.63342 | | |

# F   Broader Impacts

This work is primarily theoretical and pertains to obtaining upper bounds on the Lipschitz constant, which can serve as a measure of the robustness of deep neural networks, and does not have any direct societal impact.

