# OpenReview forum: "ECLipsE: Efficient Compositional Lipschitz Constant Estimation for Deep Neural Networks"
_NeurIPS.cc/2024/Conference — NeurIPS 2024 spotlight_

### Official Review · Reviewer_3iDD · 2024-06-13

**Soundness:** 3
**Presentation:** 2
**Contribution:** 3
**Rating:** 6
**Confidence:** 3

**Summary:**

This paper presents two novel algorithms for computing the Lipschitz constant of feedforward neural networks (NN). The starting point is a previously-known semi-definite programming (SDP) problem which enables to compute the Lipschitz constant. The paper proposes a decomposition of this SDP in sequential subproblems over layers, then relax the subproblems to enable iterative computations across layers, instead of solving a joint problem on all layers. This approach scales much better with both width and depth, as demonstrated by experiments on neural networks at initialization and after training on MNIST.

**Strengths:**

- The paper is overall well-written (see caveat in the weakness section).
- The question of computing the Lipschitz constant of neural networks is important for a number of downstream tasks. The proposed method provides estimates that are experimentally on-par with approaches based on SDP methods, while being order of magnitude faster.

**Weaknesses:**

[EDIT (Aug.7): the rebuttal answered my questions adequately. In particular, the method does provide provably-correct upper-bounds.]

- Theoretical results in Section 3.3 are a bit hard to follow, because the section gives the story behind the proposed relaxation, as well as geometric interpretation, but does not provide a main result summarizing the theoretical guarantees of the proposed approach. This is a key point, because provably correct upper-bounds on the Lipschitz constant are of course much preferable. Although it is suggested that the proposed algorithms are indeed provably correct, it is not clearly stated in the paper. So the paper would highly benefit from a clear statement on this fact, as well as a summary of the theoretical results into a theorem (see also Questions).
- The comparison with methods in the literature is limited to SDP methods, which is OK given that the main contribution is to provide a clever relaxation of these methods, but still a broader comparison would have been interesting.
- The approach only applies to feedforward NN.

**Questions:**

- I believe that Proposition 4 shows that ECLipsE-Fast provably gives an upper-bound on the true (unknown) Lipschitz constant of the neural network. Is this correct?
- Does ECLipsE also always give an upper-bound on the true Lipschitz constant? If so, are there assumptions for this to hold? If not, at which steps are approximations made? I guess this should more or less follow from Lemmas 1 and 2, and Propositions 2 and 3, but it is not clearly stated in the paper.

Minor remarks:
- Authors could consider typesetting their algorithms as “Eclipse” and “Eclipse-Fast” to improve readability.
- line 199: “such” missing.

**Limitations:**

The limitations are adequately addressed.

---

> ### Author Rebuttal · Authors · 2024-08-06
>
> Thank you for your encouraging evaluation of our work and thoughtful questions. We address all of them in detail as follows.
> >**Theoretical results in Section 3.3 are a bit hard to follow, because the section gives the story behind the proposed relaxation, as well as geometric interpretation, but does not provide a main result summarizing the theoretical guarantees of the proposed approach. This is a key point, because provably correct upper-bounds on the Lipschitz constant are of course much preferable. Although it is suggested that the proposed algorithms are indeed provably correct, it is not clearly stated in the paper. So the paper would highly benefit from a clear statement on this fact, as well as a summary of the theoretical results into a theorem (see also Questions).**
>
> Theorem 2 provides a provable guarantee for the upper-bounds on the Lipschitz constant estimates. Specifically, we show that as long as there exist $\Lambda_i>0$, $i\in \mathbb{Z}_{l-1}$, such that the inequalities in (4) hold, the Lipschitz estimate we obtain is a strict upper bound for the Lipschitz constant. Thus, the existence of positive $\Lambda_i$s and the conditions in (4) already provide the theoretical guarantee. Then, we proceed to develop two algorithms to find the  positive $\Lambda_i$s that will satisfy the provable guarantees in Theorem 2.  Finally, the theory and intuition for finding good $\Lambda_i$s are detailed in Section 3.3. We further note that the relaxation between ECLipsE and ECLipsE-Fast solely pertains to finding positive $\Lambda_i$s while trading off computational speed for accuracy, while the upper bound on the Lipschitz constant is still strict for both algorithms.
>
> Finally, as we discuss in the General Response - point III, we will move the pseudo-code of the algorithm, and additional theoretical results from the Appendix to the main text for clarity of exposition as suggested by the reviewers.
> >**The comparison with methods in the literature is limited to SDP methods, which is OK given that the main contribution is to provide a clever relaxation of these methods, but still a broader comparison would have been interesting**
>
> For non-SDP based methods, we have included a benchmark method CPLip (green dash line in Fig. 3, 4) in our experiments, which turns out to be exponentially expensive in computational cost and is therefore not scalable to the deep neural networks we consider (see Fig. 4). Of course, as the reviewer rightly points out, we do indeed focus most of our comparisons on SDP methods.
>
> >**The approach only applies to feedforward NN.**
>
> We thank the reviewer for this question. Since this was also raised by other reviewers, we have answered point I of the General Response.
>
> >**I believe that Proposition 4 shows that ECLipsE-Fast provably gives an upper-bound on the true (unknown) Lipschitz constant of the neural network. Is this correct?\
> Does ECLipsE also always give an upper-bound on the true Lipschitz constant? If so, are there assumptions for this to hold? If not, at which steps are approximations made? I guess this should more or less follow from Lemmas 1 and 2, and Propositions 2 and 3, but it is not clearly stated in the paper.**
>
> Propostion 4 states the closed form solution for $\Lambda_i$s and guarantees the positive definiteness of $M_i$ at each stage. As discussed in the first point of this review response above, the Lipschitz estimate given by ECLipsE and ECLipsE-fast are thus both provably strict upper bounds by Theorem 2.
>
> We will further clarify these points, as well as address typos and minor suggestions from the reviewer in the final version of our paper.

---

> > ### Comment · Reviewer_3iDD · 2024-08-07
> > **Thank you for the rebuttal**
> >
> > I thank the authors for their precise rebuttal, which answers adequately my questions. I raised my score accordingly.

---

> > > ### Author Response · Authors · 2024-08-07
> > >
> > > Thank you for considering our response and for increasing your evaluation. We appreciate your feedback on the paper!

---

### Official Review · Reviewer_LaVm · 2024-07-03

**Soundness:** 3
**Presentation:** 3
**Contribution:** 3
**Rating:** 6
**Confidence:** 3

**Summary:**

This paper tackles the problem of computing the Lipschitz constant of a neural network. Since computing the exact Lipschitz constant is NP-hard, efforts have been made to obtain tight upper bounds on the Lipschitz constant. This paper builds on the work of LipSDP [1], which involves solving a large matrix verification problem. Since that the large matrix verification problem grows significantly for both deeper and wider networks, this paper proposes a compositional approach to estimate the Lipschitz constants of deep feed-forward neural networks more efficiently. First, the authors obtain an exact decomposition of the large matrix verification problem into smaller sub-problems and, then, exploiting the underlying cascade structure of the network the authors develop two algorithms to compute a bound on the Lipschitz constant:
- The first algorithm explores the geometric features of the problem and provides a tight estimate of the Lipschitz constant by solving small semidefinite programs (SDPs) that are only as large as the size of each layer.
- The second algorithm relaxes these subproblems and provides a closed-form solution to each subproblem for extremely fast estimation, altogether eliminating the need to solve SDPs altogether.

Finally, the authors provide extensive experiments to show the different levels of tradeoffs between efficiency and accuracy of the two algorithms. They show that their approach provides a steep reduction in computation time while yielding Lipschitz bounds that are very close to, or even better than, those achieved by state-of-the-art approaches.

**Strengths:**

- The paper is clear and well written. The problem of providing a scalable algorithm for computing the Lipschitz of neural networks is interesting and important.
- The exact decomposition of the large matrix verification problem into smaller subproblems is very interesting.
- The two algorithms for computing the sequence of inequalities provide interesting trade-offs. The first algorithm (ECLipsE) looks, if I understand correctly, like a direct improvement of LipSDP, since ECLipsE provides the same value as LipSDP in a more efficient way.
- The second algorithm also looks interesting as it provides a way to compute the Lipschitz constant without SDPs.

**Weaknesses:**

- it looks like the approach is restricted to a very limited set of neural networks (feedforward neural networks), can the approach be used for convolutional neural networks?
- In the experiments, the authors use randomly generated neural networks for their first set of experiments, in my experience it is usually easier to compute SDP on random weight matrices than on trained weight matrices due to conditioning, could the authors provide results of these experiments with trained networks?
- Can the authors clarify if ECLipsE has to compute all subproblems at the same time or if a sequential approach is possible?
- I assume that the ECLipsE algorithm uses Matlab for SPD optimization, have the authors tried using a deep learning framework (e.g. PyTorch) for ECLipsE-Fast? Could ECLipsE-Fast be used during training, e.g. for regularization?

**Questions:**

See Weaknesses

---

> ### Author Rebuttal · Authors · 2024-08-06
>
> Thank you for your encouraging evaluation of our work and thoughtful questions. We address all of them in detail as follows.
> >**It looks like the approach is restricted to a very limited set of neural networks (feedforward neural networks), can the approach be used for convolutional neural networks?**
>
> Since this was also raised by other reviewers, we have answered it in point I of the General Response.
> >**In the experiments, the authors use randomly generated neural networks for their first set of experiments, in my experience it is usually easier to compute SDP on random weight matrices than on trained weight matrices due to conditioning, could the authors provide results of these experiments with trained networks？**
>
> We do in fact have experiments for both cases. Section 4.1 considers randomly generated neural networks and Section 4.2 considers neural networks trained for  MNIST tasks. Note that our algorithms estimate the Lipschitz constant for a given model, that is, all the parameters are fixed and the algorithms are implemented out after the model is trained. Therefore, we did not observe any additional difficulties in estimating the Lipschitz constant for trained weight matrices. Please see Section 4.2 for detailed results on trained weight matrices.
> >**Can the authors clarify if ECLipsE has to compute all subproblems at the same time or if a sequential approach is possible?**
>
> ECLipsE computes the subproblems by sequence. Starting with $i=1$, the SDP problem as expressed in (6) requires the information matrix $M_{i-1}$ passed on from the computation at the $(i-1)$-th stage. Therefore, it is a sequential approach, aligning with the cascaded neural network structure.
> >**I assume that the ECLipsE algorithm uses Matlab for SPD optimization, have the authors tried using a deep learning framework (e.g. PyTorch) for ECLipsE-Fast? Could ECLipsE-Fast be used during training, e.g. for regularization?**
>
> Yes, ECLipsE uses Matlab with solver Mosek to solve the SDPs. For ECLipsE-Fast, the solutions for $\Lambda_i$s at each step are actually obtained in closed-form as stated in Proposition 4, and at the last step the Lipschitz estimate is also in closed-form as given in Proposition 1. Thus, there is no need to solve any SDPs at all for ECLipsE-Fast; we obtain fast estimates by only evaluating closed-form analytical expressions without the need to use any deep learning framework. Lastly, we believe our efficient, scalable and accurate methods like ECLipsE-Fast will facilitate robust training for neural networks in the future, although this is beyond the scope of the present paper.

---

> > ### Comment · Reviewer_LaVm · 2024-08-08
> >
> > Thank you for the clarifications. I agree that the unrolling approach could be used in the context of this paper - although it might not be the most scalable approach (unrolling a large convolution leads to extremely large matrices). I agree that the exploration of other types of architectures could be left to future work, and I am in favor of accepting this paper and will raise my score accordingly.

---

> > > ### Author Response · Authors · 2024-08-08
> > >
> > > Thank you for considering our response, and raising your evaluation! We also appreciate your feedback on improving our manuscript!
> > > We agree that unrolling the convolutional layer and applying FNN based methods may not be the most practical solution. Further study is necessary to develop scalable algorithms for other network architectures, which will be the subject of future work.

---

### Official Review · Reviewer_kXCT · 2024-07-11

**Soundness:** 3
**Presentation:** 3
**Contribution:** 3
**Rating:** 6
**Confidence:** 4

**Summary:**

The paper proposes two algorithms, ECLipsE and ECLipsE-Fast, to estimate the Lipschitz constant of a feed-forward neural network. The estimation of the Lipschitzness plays a crucial role in certifying the robustness of neural networks and is known to be an NP-hard problem. The proposed algorithms are based on the LipSDP of Fazlyab et al. (2019), which describes the semidefinite program (SDP) that an upper bound of such a Lipschitz constant should generally satisfy. The authors decompose the original large SDP into smaller layer-wise SDPs to improve the scalability of the original approach. The validity of the resulting methods, ECLipsE and ECLipsE-Fast, is supported by theoretical analyses. Experiments show a steep reduction in computation time while maintaining competitive accuracy compared to the LipSDP.

**Strengths:**

Originality: The proposed algorithm is novel and clearly distinguishes itself from prior works.

Quality: The motivation of the work is clearly stated and explained. Prior works are also well-discussed. The authors provide thorough and sound mathematical justification and geometrical intuitions for the two algorithms.

Clarity: The paper is well-written including the methodology, and the motivation is clear.

Significance: Compared to LipSDP, the proposed algorithm provides significant improvement in terms of efficiency and addresses concerns stated in the motivations of the paper in the beginning.

Overall, I feel this is a good paper with a promising approach equipped with a thorough and interesting mathematical justification.

**Weaknesses:**

Overall, in my view, the main weakness of this paper is that there is no explicit comparison with other works trying to improve the scalability of LipSDP (such as [20]). As a result, while the paper indeed improves the original LipSDP in a new way, it is unclear how significant this work is taking into account existing literature. In addition, the limitations should be more carefully discussed. See below for further detailed comments and advice.

### **Quality**

(W-Q1) There are some typos: l.60 (constans), p.3 eqation (3) (index of the bottom right element is i+1 but should be l), l.172 (in the matrix WMW there are 2 unnecessary “L”), l.264 (computatinoal), between l.420 and l.421 (i\in \mathbb{R}^n), p.13 equation (18) (if i=0 should be if i=1?), l.477 (functionsare), l.480 (the is norm),

(W-Q2) I feel that some use of words is misleading.

1. l.60 “We develop a sequential Cholesky decomposition technique to obtain […]“: If this is about Theorem 2, it directly uses the result of Agarwal et al., so maybe “use, employ” would be better than “develop”.

2. l.10 “The first algorithm [...] enables us to provide a *tight* Lipschitz constant”, l.56 “ algorithm [...] enables us to provide an *accurate* Lipschitz”... What do you mean by “tight” and “accurate”? Since the proposed algorithms are using some simplifications, I think that those adjectives should be relative, i.e., only used in comparison with something else. Notably, the experiments (e.g., Figure 3) show that CPLip is far more precise than the proposed algorithms so the authors should clarify the meaning of “tight” and “accurate” (or delete them if there is no justification) when describing their own algorithms.

3. Between l.462 and l.463, you write “$N/ci(M_i)^{-1}\ge0$”. Semi-positive definiteness was only defined for symmetric matrices but this one is not necessarily symmetric. There should be an easy fix, but it is ambiguous what you mean by this sign.

(W-Q3) In Proposition 3, the authors simplify Theorem 2 by setting $M_i$ to $c_iW_{i+1}^\top  W_{i+1}+N$ without any proper discussion about this choice (Q2).

(W-Q4) The proof of positive definiteness of $M_i$ is missing in Proposition 3 (Q3).

(W-Q5) (minor) Some references should be adjusted: [2] was accepted at ICLR2018, some capital letters are missing (L of lipschitz)...

### **Clarity**

(W-C1) Some parts may require clarifications (See also Questions):

1. (minor) l.111 “[30] provides a counterexample to the most accurate approach”: The concrete property of the most accurate approach disproved by [30] could be explained in a few words.
2. (minor) A mathematical comparison of the computational complexity between your approach and LipSDP will largely help the reader to quickly understand the difference in scalability.

(W-C2) While limitations of proposed algorithms are all stated, they are dispersed throughout the paper. A short subsection summarizing them would be useful. (See also Limitations)

(W-C3) It was a bit difficult to understand the idea of the proposed algorithms from Subsection 3.2. Perhaps showing Algorithm 1 in the main text would be better.

### **Significance**

(W-S1) (major) One of the main contributions of this work is the scalability of the proposed algorithms. However, the authors do not compare their method with other accelerations of LipSDP. Therefore, it is difficult to situate their work within the broader context of efforts to improve the scalability of LipSDP. At least, comparing the algorithms with that of Wang et al. (2024) [20] is important to clarify these points.

(W-S2) The algorithms were run on medium-scale neural networks, and it is difficult to imagine the scalability of ECLipsE and ECLipsE-Fast. Experiments with even larger architectures (for example, those for training on CIFAR or ImageNet) would be more convincing.

(W-S3) I feel experiments of Subsection 4.2 are a little bit redundant (Q4).

(W-S4) Limitations of the algorithms should be discussed in more detail and more explicitly.

**Questions:**

Q1: Is Theorem 1 *necessary* and sufficient? If not, this should be added to the limitations of the work.

Q2: Why did the authors set $M_i$ to $c_iW_{i+1}^\top  W_{i+1}+N$ in Proposition 3?

Q3: How is the positive definiteness of $M_i$ guaranteed in Equation (6) and Proposition 3?

Q4: What was the motivation to run experiments of Subsection 4.2?

**Limitations:**

There are several limitations of the algorithms that are worth mentioning:

1. The algorithms (Lemma 1) need that the last weight matrix is full row rank. So, we cannot blindly apply them to any feed-forward neural network.

2. There is a simplification when transforming Theorem 2 into Proposition 3 by limiting the expression of  $M_i $ to $c_iW_{i+1}^\top  W_{i+1}+N$. This may lead to looser bounds than the original LipSDP.

3. The algorithms cannot be applied to CNNs and residual networks.

---

> ### Author Rebuttal · Authors · 2024-08-06
>
> We thank the reviewer for a thorough reading of our manuscript and providing several suggestions to improve the clarity of our presentation (see General Response - point III). We individually address the technical questions raised by the reviewer below, and provide additional experiments benchmarking our algorithms (results in **attached PDF** of the General Response and discussion below).
> >**One of the main contributions of this work is the scalability of the proposed algorithms. [..] At least, comparing the algorithms with [..] [20] is important.**
>
> While [20] was too recent to reproduce at the time of our submission (<1 month), we can now provide additional experiments using their open-source code and considering the same NNs in Section 4.1 Case 1 and Case 2. The results in the **attached PDF** (in the General Response) demonstrate that (i) ECLipsE and [20] have similar computation times for smaller networks; however, the computation time for [20] grows more rapidly for both deeper and wider networks, and (ii) ECLipsE-Fast remains orders of magnitude faster than all algorithms while providing Lipschitz estimates that are very close to those achieved by LipSDP-Layer, and (iii) **importantly**, [20] provides inadmissible estimates for moderate networks, returning as much as $10^4-10^6$ times and 10-100 times the trivial bound in Tables 2a and 1a respectively. Note that all the estimates are normalized with respect to trivial upper bounds.
>
> Another work on improving the scalability of LipSDP is [33]. However, as acknowledged by the authors in their footnote, their acceleration depends on LipSDP-Network, which is proved to be invalid by [30]. Therefore, we do not include it as a benchmark.
>
> >**The proof of positive definiteness of $M_i$ is missing in Proposition 3.**
>
> Thank you for raising this question.
> The first part of the proof of Proposition 3 (l.463-465) showing the non-emptiness of the feasible region implies the positive definiteness of $M_i$ for the next stage. This is because $M_i=\Lambda_i-\frac{1}{4}\Lambda_iW_i(M_{i-1})^{-1}W_i^T\Lambda_i$. From (10), $M_i>c_iW_{i+1}^TW_{i+1}>0$ as $c_i>0$ and $W_{i+1}^TW_{i+1}\geq 0$. Then, with $M_0>0$, $M_i$ is guaranteed to be positive definite at each step. We will supplement the proof for clarity.
>
> >**In Proposition 3, the authors simplify Theorem 2 by setting $M_i$ to $c_iW_{i+1}^TW_{i+1}+N$ without any proper discussion about this choice.**
>
> There is no simplification here, and we can write $M_i$ **exactly** as $c_iW_{i+1}^TW_{i+1}+N$. We briefly summarize the proof here.
> Since $M_i$ is positive definite (see response above), and $W_{i+1}^TW_{i+1}$ is positive semidefinite, there exists constant $C$ such that for any $c\in[0,C]$, $M_i-cW_{i+1}^TW_{i+1}\geq 0$. Now, for the $i$-th layer, let $c_i$ be the largest possible $C$ such that $M_i-cW_{i+1}^TW_{i+1}\geq 0$ holds. Then, $N=M_i-c_iW_{i+1}^TW_{i+1}$ is a positive semidefinite matrix and also a singular matrix. We show this by contradiction in the proof of Proposition 3; see lines 465-468, starting at `we now prove by contradiction..'. (we do not include this here for brevity). Therefore, we can equivalently write $M_i=c_iW_{i+1}^TW_{i+1}+N$, where $N$ is a positive semidefinite matrix and also a singular matrix.
> >**The algorithms were run on medium-scale neural networks, and it is difficult to imagine the scalability of ECLipsE and ECLipsE-Fast [..]**
>
> ECLipsE-Fast has closed-form solution and is therefore scalable, no matter the network size. We did not present larger networks because the existing benchmarks already exceed the cutoff time of 30 min for medium-sized networks. Further, from existing experiments, our acceleration is already pronounced with promising accuracy, and the advantage will be even more significant for larger networks.
> >**What was the motivation to run experiments of Subsection 4.2?**
>
> The motivation for Section 4.2 is two-fold. First, we show that our algorithms apply not only to randomly generated weights but also to those trained for some specific tasks. Second, although ECLipsE provides a tighter estimate in general, we show an interesting case where ECLipsE-Fast is more favorable compared to ECLipsE (with similar accuracy but much faster speed).
> >**[..] need that the last weight matrix is full row rank [..] cannot blindly apply them to any feed-forward neural network.**
>
> Since this was also raised by another reviewer, we address it in point II of the General Response. In short, this assumption is not necessary, and both algorithms apply even when it is not satisfied.
> >**[..] cannot be applied to CNNs and residual networks.**
>
> Since this question was also raised by other reviewers, we have answered it under General Response - point I.
> >**(a) l.10 [..] What do you mean by “tight” and “accurate”? \
> (b) Between l.462 and l.463, you write “$N/c_i(M_i)^{-1}\geq 0$
> ”. Semi-positive definiteness was only defined for symmetric matrices but this one is not necessarily symmetric [..]**
>
> (a) In the Introduction, "tight" and "accurate" are used in a general sense, to express that our method provides Lipschitz estimates that are comparable to existing methods while achieving significantly enhanced scalability. We will rephrase in response to the reviewer's suggestions.
>
> (b) By the definitions of $N$ and $M_i$, they are indeed guaranteed to be symmetric.
> >**(minor) (a) l.111 “[..] the most accurate approach disproved by [30] could be explained [..]\
> (b) A mathematical comparison of the computational complexity between [..] LipSDP will largely help [..]**
>
> (a) [30] gives a counterexample showing that the Lipschitz estimate from LipSDP-Network is not a strict upper bound.
>
> (b) Please see point (1) of our response to Reviewer kbMy for a computational complexity analysis (not repeated here due to space constraints).

---

> > ### Comment · Reviewer_kXCT · 2024-08-08
> >
> > Thank you for these clarifications and additional experiments. The mathematical justification of the algorithms now seems valid to me. As you suggested, a clearer explanation of the points in the proof I asked for clarification on may be beneficial for the updated version of your paper. I am in favor of accepting this paper and will raise my score accordingly.
> >
> > Still just one detail:
> >
> > >(b) By the definitions of $N$ and $M_i$, they are indeed guaranteed to be symmetric.
> >
> > I agree that $N$ and $M_i$ are both symmetric, but the product of symmetric matrices (e.g., $N/c_i (M_i)^{-1}$) is not necessarily symmetric. That is why I pointed out that $N/c_i (M_i)^{-1}\geq 0$ was not well-defined as $\geq$ was only defined for symmetric matrices in the Notation. Shifting the focus of the discussion in l.462-463 on the singular values without using $\geq 0$ should solve the problem anyway.

---

> > > ### Author Response · Authors · 2024-08-08
> > >
> > > We thank you for considering our response, and for raising your evaluation. We are truly grateful for the thorough and detailed suggestions on improving our manuscript! Yes, the product of two symmetric matrices is not necessarily symmetric. We can instead focus on the discussing eigenvalues of $N(M_i)^{-1}$ and prove that it can only have non-negative eigenvalues. We will edit the proof accordingly in our final version.

---

### Official Review · Reviewer_kbMy · 2024-07-14

**Soundness:** 3
**Presentation:** 2
**Contribution:** 3
**Rating:** 7
**Confidence:** 4

**Summary:**

The paper proposes two novel Lipschitz constant estimation algorithms ECLipsE and ECLipsE-Fast. They are supported by a new decomposition theory developed for the Lip-SDP framework, derived by applying an existing theory (Lemma 2 of [31]). Experiments demonstrate the estimation accuracy and acceleration using toy radom networks and networks trained on MNIST data, by comparing with classical Lipschitz constant estimation method.

**Strengths:**

The targeted research problem is important and useful.

The decomposition theory is new and the two estimation algorithms are novel. I highly appreciate the beauty of the application of Lemma 2 of [31] in the proposed theory development.

The achieved result improvement is satisfactory for deep networks.

**Weaknesses:**

(1) The proposed algorithm is efficient at addressing network depth, but does not look at the width.  The theory behind self-explains the success of its capability of handling depth.  However, layers with very high numbers of neurons still pose challenges for the sub-problems, e.g., solving for Eq. (6) and Eq. (7). There is a lack of mentioning of this. Also the experiments only studied some modest widths up to 100 neurons. It would be good to see more empirical results with higher neuron numbers in each layer,  to understand the limit of the proposed algorithms on network width.

(2) In experiments, there seems a missing comparison with the “parallel implementation by splitting” version of Lip-SDP as reported in their paper [14], which was proposed to address the depth issue.

(3) The  method description can be improved, e.g., being more organised. The paper can present information that is more important and helpful to practitioners and general readers’ understanding in main paper, while leave some analysis and supporting lemmas to appendix. Personally, I find it helpful to see in the main paper a description of the existing Lemma 2 in [31] and the pseudo code of the proposed algorithm.

**Questions:**

The authors are invited to address my comments (1) and (2) in weakness section. If possible, it would be good to see some added results to help understand more the width capacity of the proposed algorithms.

**Limitations:**

Discussion on discussion is pretty limited. For instance, it  can be improved around the network width issue.

---

> ### Author Rebuttal · Authors · 2024-08-06
>
> Thank you for your encouraging evaluation of our work and for the insightful questions on our experiments. We address all your questions as follows, and provide additional experiments to show the strength of our method for wide networks.
>
> >**(I). The proposed algorithm is efficient at addressing network depth, but does not look at the width. The theory behind self-explains the success of its capability of handling depth. However, layers with very high numbers of neurons still pose challenges for the sub-problems, e.g., solving for Eq. (6) and Eq. (7). There is a lack of mentioning of this. (II). Also the experiments only studied some modest widths up to 100 neurons. It would be good to see more empirical results with higher neuron numbers in each layer, to understand the limit of the proposed algorithms on network width. (III). Discussion on discussion is pretty limited. For instance, it can be improved around the network width issue.**
>
> We thank the reviewer for these important questions, and answer them here. While we acknowledge that our computational advantage is more pronounced with respect to network depth, the speedup for wide networks is also significant, especially when the  network is also deep. To see this, we can assess the computational complexity for ECLipsE and LipSDP-Neuron. Suppose a neural network has $n$ hidden layers with $m$ neurons. Then, the large matrix in Theorem 1 has dimension $nm+O(1)$ and the decision variable is of size $nm+O(1)$. The computational complexity for solving an LMI with decision variables of matrix size $A$ and $B$ is $O(A^3+A^2B^2)$. Therefore, the computational cost for LipSDP (solving SDP involving the large matrix) is $O((nm+O(1))^3+(nm+O(1))^2(nm+O(1))^2)=O(n^4m^4)$. Contrarily, ECLipsE solves $n$ sub-problems as in Eq. (6), with each involving matrix of size $O(m)$ and $m$ decision variables. The corresponding total computational cost is $n\times (O(m^3+m^2m^2))=O(nm^4)$. We can see that the complexity is significantly decreased in terms of the depth, but is the same in terms of the width, immediately indicating the advantage for deep networks. Nevertheless, as $m$ grows, the difference between $O(n^4m^4)$ and $O(nm^4)$ are still enlarged drastically, especially with large $n$. More importantly, for ECLipsE-Fast, we note that we  do not need to solve any SDPs as the solutions are all provided in closed-form from Propositions 1 and 4. Thus, the computational cost drops to $n\times O(m^3)=O(nm^3)$. This is the fastest one can expect if the weights on each layer are treated as a whole. Admittedly, if a neural network is considerably wide, it can still pose challenges to the sub-problems regardless of all the accelerations we have achieved, in which case we will have to apply methods that split the weights themselves, introducing some unavoidable conservativeness.
>
> From the experiment side, we initially considered networks with up to 100 neurons, since benchmark methods like LipSDP-Neuron with 50 neurons and LipSDP-Layer with 70 neurons already fail to return estimates within the cutoff time of 15 min (see Figure 4). Meanwhile, both ECLipsE and ECLipsE-Fast still work well in these settings, demonstrating our advantages regarding width.
>
> To further illustrate the strengths and limitations, we consider  randomly generated NNs with 50 layers as shown below, and find that (i) ECLipsE-Fast is extremely fast even for very wide networks, with a running time of only 15.63 seconds  for a width of 1000, while the computation time for LipSDP-Layer grows significantly, and (ii) ECLipsE is comparable to LipSDP-Neuron split into 5 sub-networks in terms of time performance (note that LipSDP-Neuron cannot return estimates for any of the cases without splitting, which slightly decreases its accuracy with respect to ECLipsE). We will include these additional results and discussions in the final version. Note that all the estimates are normalized with respect to trivial upper bounds.
>
> **Normalized Lipschitz Estimates for Randomly Generated Neural Network with 50 Layers**
> | Neuron | ECLipsE  | ECLipsE-Fast | LipSDP-Neuron Split by 5 | LipSDP-Layer Split by 5 |
> |--------|-------|--------------|--------------|----------------------|
> |150|0.743745|0.867548|0.758217| 0.87342|
> |200| 0.773494 | 0.883758|0.785171| 0.888306|
> |300| >30min|0.897008| >30min|0.899164|
> |400|   |0.899916|       | >30min|
> |500|    |0.903529|   |  |
> |1000|   |0.912093|   |    |
>
> **Time Used for Randomly Generated Neural Network with 50 Layers (Seconds)**
> | Neuron | ECLipsE | ECLipsE-Fast | LipSDP-Neuron Split by 5 | LipSDP-Layer Split by 5 |
> |--------|---------|--------------|--------------------------|-------------------------|
> | 150 | 387.7| 0.387262 | 451.07 | 93.129 |
> | 200 | 1386.6| 0.584115 | 1377.9  | 210.16 |
> | 300 | >30min| 1.321177 | >30min | 612.47     |
> | 400 |   | 2.657505 |  | 2110.9 |
> | 500 |  | 3.7435 |  | >30min |
> | 1000 |  | 15.63342 | |  |
>
> >**In experiments, there seems a missing comparison with the “parallel implementation by splitting” version of Lip-SDP [...]  to address the depth issue.**
>
> The ``parallel implementation by splitting'' version of LipSDP is implemented in Section 4.1 Case 4.3 directly using the code provided by [32], where we compare 3 ways of splitting, namely into 3, 5, and 10 layers respectively. The results are promising as discussed in Case 4.3: ECLipsE-Fast is the fastest algorithm and outperforms LipSDP-Layer regardless of how we split the neural networks. ECLipsE is also shown to be relatively more accurate and efficient than all LipSDP methods, no matter the split.
> >**The method description can be improved, e.g., being more organised [...] I find it helpful to see in the main paper a description of the existing Lemma 2 in [31] and the pseudo code of the proposed algorithm.**
>
> We thank the reviewer for their suggestions on improving the clarity of our paper, and will revise accordingly - please see General Response point III.

---

> > ### Comment · Reviewer_kbMy · 2024-08-11
> >
> > I thank the authors for their very clear explanation on their algorithm complexity with respect to neural network depth and width, and providing experiments with higher width to demonstrate algorithm capacity, while acknowledging the limit/boundary. I am happy to see the paper to be accepted, therefore will increase my score to 7. Meanwhile, I recommend the authors to discuss around "width" in their discussion/limitation section.

---

> > > ### Author Response · Authors · 2024-08-12
> > >
> > > We thank you for your feedback on our manuscript, particularly on wider vs deeper networks, and for raising your score! We will add the experiments on width and a discussion section on limitations.

---

### Official Review · Reviewer_qsZt · 2024-07-17

**Soundness:** 4
**Presentation:** 3
**Contribution:** 4
**Rating:** 8
**Confidence:** 4

**Summary:**

The authors are able to decompose a particular case of LipSDP-neuron exactly into a series of sub-problems leading to the proposed algorithm ECLipSE. In the case of the relaxed LipSDP-layer, it can be shown that each sub-problem can be solved analytically and eliminate the need for solving an SDP. The proposed algorithm ECLipSE-fast can provide Lipschitz estimates of deep-NN quite fast at the expense of increased conservativeness.

**Strengths:**

-The paper is well written and the theoretical arguments are well motivated and connected to the the numerical experiments. The paper mainly builds upon LipSDP, but I believe the theoretical insights to be novel.

-The insight that LipSDP may be simplified into sub-problems is a significant contribution and advances the practical value of LipSDP. In particular, the proposal of ECLipsE-Fast which is an analytical solution to a relaxed sub-problem shows a significant improvement over the previous naive product bound and is very scalable.

**Weaknesses:**

-It seems that the approach doesn’t yet apply to residual networks or CNN commonly found in state-of-the-art vision models. For this reason, it seems difficult to show improved certified robustness on common image classification benchmarks which would benefit the most from increased scalability (CIFAR10-100, Tiny-imagenet, etc).

-Certified robustness is mentioned as an application at several points in the paper. While the scalable and tighter Lipschitz estimates on random networks and MNIST certainly suggest some improvements, no practical measures of certified robustness are presented (e.g. robust accuracy).  For certified robust accuracy on MNIST, it is common to use 1-Lipschitz parameterization (SLL, AOL, other direct parameterizations) which eliminates the need for Lipschitz estimation. Would applying ECLipSE to composed 1-Lipschitz networks provide significantly tighter estimates in this case?

**Questions:**

-How does the full row rank assumption limit the applications of ECLipSE’s? Is it possible to relax this assumption? It seems like networks considered in this paper are only of constant width or decreasing in output dimension in the case of MNIST which will usually satisfy the full row-rank assumption.

-Can you comment on the slight gap between the Lipschitz constant estimates shown in Appendix E between ECLipsE and LipSDP-Neuron? Is it that LipSDP is not as accurate in larger settings or is ECLipsE somehow slightly conservative?

**Limitations:**

Limitations are adequately addressed.

---

> ### Author Rebuttal · Authors · 2024-08-06
>
> Thank you for your insightful comments and positive evaluation of our work. We address your concerns as follows.
> >**It seems that the approach doesn't yet apply to residual networks or CNN commonly found in state-of-the-art vision models. For this reason, it seems difficult to show improved certified robustness on common image classification benchmarks which would benefit the most from increased scalability (CIFAR10-100, Tiny-imagenet, etc).**
>
> We thank the reviewer for this question; since this was also raised by other reviewers, we have answered point I of the General Response.
> >**Certified robustness is mentioned as an application at several points in the paper. While the scalable and tighter Lipschitz estimates on random networks and MNIST certainly suggest some improvements, no practical measures of certified robustness are presented (e.g. robust accuracy).**
>
> We show in experiments that for both randomly generalized networks, as well as ones trained for specific tasks, our algorithms consistently give promising results. Thus, we believe our method will benefit studies that certify robustness by building a relationship between the Lipschitz constant and robustness metrics (e.g. robust accuracy). For instance, [19] builds Lipschitz-based surrogates for certified radius, which is a classical robustness measure for classification tasks. We believe that such applications will benefit from our efficient Lipschitz estimation algorithms.
> >**For certified robust accuracy on MNIST, it is common to use 1-Lipschitz parameterization (SLL, AOL, other direct parameterizations) which eliminates the need for Lipschitz estimation. Would applying ECLipSE to composed 1-Lipschitz networks provide significantly tighter estimates in this case?**
>
> We thank the reviewer for this question. While we consider a very general FNN structure, with only a slope restrictedness assumption on the activation functions, we anticipate that incorporating side information corresponding to specific parametrizations will yield better estimates in any algorithm. However, this is the subject of future work. Moreover, to the best of our knowledge, there is no theoretical guarantee of obtaining a tighter estimate for 1-Lipschitz networks.
>
> We now turn to the question of whether Lipschitz constant estimation is necessary, given developments like 1-Lipschitz networks. While 1-Lipschitz parameterization is commonly used in robust training and the robustness is guaranteed by way of parameterization, the 1-Lipschitz parameterization has limited expressive power. For example, AOLs restrict the output to be a sum of individual contributions from inputs, which may not be sufficient for some complex tasks. In contrast, our theory applies to FNNs, which are a very general network structures with universal expressive power adopted in various network designs. Moreover, our theoretical development only requires the rather mild assumption of slope-restrictedness of the activation function, which is satisfied for most cases. Thus, our work facilitates fast Lipschitz constant estimation for more general structures, while still being applicable to direct parametrizations.
> > **How does the full row rank assumption limit the applications of ECLipSE’s? Is it possible to relax this assumption? It seems like networks considered in this paper are only of constant width or decreasing in output dimension in the case of MNIST which will usually satisfy the full row-rank assumption.**
>
> We thank the reviewer for this question. Since this was also raised by another reviewer, we have answered it under the General Response section - see part II therein for a detailed response. In short, this assumption is not necessary, and both algorithms apply even when it is not satisfied.
> >**Can you comment on the slight gap between the Lipschitz constant estimates shown in Appendix E between ECLipsE and LipSDP-Neuron? Is it that LipSDP is not as accurate in larger settings or is ECLipsE somehow slightly conservative?**
>
> The slight gap in performance, in fact, exhibits the high accuracy of ECLipsE.  While LipSDP-Neuron provides en efficient approach to estimate Lipschitz constants, it is practically not as scalable as our algorithms for large networks, and yields unacceptably long running times (>15min for 50 layers with only 60 neurons). Therefore, to implement LipSDP-Neuron, as suggested in [14], the NN must be split into several small sub-networks. However, the different sub-networks are treated independently, and their Lipschitz constants are multiplied at the end, thus completely cutting off the relationship among sub-networks. In contrast, our method always keeps the information from previous layers due to the exact decomposition in Theorem 2. This explains why LipSDP-Neuron yields less accurate results compared to EClipsE for the large networks discussed in Section 4.1 Case 3, and is precisely where our advantages lie.
>
> Theoretically, LipSDP is a centralized method where solving a large matrix SDP is unavoidable. In this context, we note that ECLipsE, being a distributed algorithm, will naturally result in a trade-off between speed and accuracy, yielding more conservative estimates than centralized algorithms like LipSDP. However, as we demonstrate in our experiments, the Lipschitz constant estimates from both ECLipsE and ECLipsE-Fast are fairly close to those obtained using LipSDP, while providing a significant computational advantage.

---

> > ### Comment · Reviewer_qsZt · 2024-08-12
> >
> > Thank you for addressing my concerns about the row-rank assumption and the gap of the estimates of Appendix E. I think ECLipsE is an interesting result with high impact in the neural-network robustness community. I have raised my score.

---

> > > ### Author Response · Authors · 2024-08-13
> > >
> > > Thank you for your positive evaluation of our work, and for raising your score! We appreciate your feedback on the paper!

---

### Author Rebuttal · Authors · 2024-08-06

We are extremely grateful to the reviewers for their detailed, thorough, and constructive feedback. We are glad to read that the reviewers found paper to be interesting, novel, practical and well-written. We appreciate the suggestions from Reviewers kbMy, 3iDD on enhancing the clarity of writing, Reviewer kXCT on providing additional experiments benchmarking to the state-of-the-art, and Reviewers qsZt, kXCT, LaVm, 3iDD on the generalization of our algorithms to other neural network architectures. We address all the reviewers' concerns and questions in individual responses, and provide additional  experiments to illustrate the strength of our methods. First, we address common questions raised by multiple reviewers and ones that provide additional benchmarking experiments here.

**I. On the applicability of our algorithms to other neural network architectures such as CNNs (Reviewer qsZt, Reviewer kXCT, Reviewer LaVm, Reviewer 3iDD):**

Several reviewers raise the question of whether our algorithms are applicable beyond feedforward neural networks (FNNs) to other classes such as convolutional neural networks (CNNs) and residual networks. While our work exploits the mathematical structure of the underlying matrices arising from cascaded architectures to develop fast algorithms, the applicability of our algorithms is not restricted to only FNNs. In the case of CNNs, we can adopt a strategy similar to LipSDP where the CNN can be unrolled into a large fully connected neural network, following which we can apply both ECLipsE and ECLipsE-Fast.

While our current study focuses on significantly accelerating the computation of Lipschitz constants for FNNs, future work will involve exploring the mathematical structures of other architectures such as residual networks to develop similarly fast algorithms for Lipschitz constant estimation.

**II. On the full row rank assumption on the last weight matrix in Lemma 1 (Reviewer qsZt, Reviewer kXCT):**

We thank the reviewers for this insightful question. First, we would like to clarify that both ECLipsE and ECLipsE-Fast are still valid even if the full row rank assumption is not satisfied. This is due to the fact that at the last stage, the Lipschitz estimate is in fact given by a closed-form expression with no requirement on the row rank of $W_l$ as Proposition 1. We note that the reason that this assumption was made is solely for ease of exposition of the intuition arising from the geometric features of the problem. Specifically, as discussed after Lemma 2, if the weight matrix is full ranked, then minimizing $\sigma_{max}(F_i)$ aligns with minimizing $\sigma_{max}\left(W_l^TW_l(M_{l-1})^{-1}\right)=\sigma_{max}\left(W_l(M_{l-1})^{-1}W_l^T\right)=\sigma_{max}(F_{l})$ at the last stage, from a geometric perspective. However, we note that the algorithms themselves do not rely on this fact due to the closed-form expression in Proposition 1.

Also, practically speaking, it is common to set the dimension of the last hidden layer to be much larger than the output dimension. Thus, $W_l$, as a fat matrix, is almost always full row ranked (see the discussions after Lemma 1).

We acknowledge that this choice of presentation may have caused some ambiguity regarding the necessity of this assumption. We will clarify this in the final version of the paper.

**III. On the organization of the algorithm description in the paper (Reviewer kbMy, Reviewer kXCT, Reviewer 3iDD):**

Several reviewers have suggested moving the existing Lemma 2 in [31] and the pseudo code of the proposed algorithm to the main paper for better organization and presentation of the theoretical results in the paper. We thank the reviewers for this suggestion. Given the additional page available for the camera-ready version, we will incorporate them into the main body of the final paper. We will also fix all typos and take into account editorial suggestions from various reviewers in the final version.

**IV. Additional Benchmarking Experiments (Reviewer kXCT):**

Reviewer kXCT suggested that our algorithms should be benchmarked against those in [20]. While [20] was too recent to reproduce at the time of our submission (<1 month), we now provide additional experiments (see attached PDF)
benchmarking our algorithms with respect to [20], considering the same NNs in Section 4.1 Case 1 and Case 2. In short, (i) ECLipsE and [20] have similar computation times for smaller networks; however, the computation time for [20] grows more rapidly for both deeper and wider networks, and (ii) ECLipsE-Fast remains several orders of magnitude faster than all algorithms while providing Lipschitz estimates that are very close to those achieved by LipSDP-Layer, and (iii) **importantly**, [20] provides inadmissible estimates for moderate networks, returning as much as $10^4-10^6$ times and 10-100 times the trivial bound in Tables 2a and 1a respectively.

---

### Decision · Program_Chairs · 2024-09-25

**Decision:**

Accept (spotlight)

**Comment:**

This paper proposes a novel, efficient algorithm to estimate the Lipschitz constant of neural networks. The algorithm is based on SDP based methods, developed to estimate the Lipschitz constant. The main idea is breaking the SDP into sequential subproblems which involve matrix inequality on smaller matrices.  Using this decomposition, the paper proposes two algorithms with different computational cost and accuracy, allowing to establish cost-accuracy trade-offs. In particular, the proposed method significantly reduces the computational complexity with depth. Experiments show that this proposed method can achieve almost the same estimate while being scalable to deeper networks. The paper is well-motivated, well-written with a solid contribution. While the proposed method is limited to feedforward networks, this limitation is included in the discussion. Thus, I recommend to accept this paper for a spotlight presentation.